# Innovation and constraint leading to complex multicellularity in the Ascomycota

Tu Anh Nguyen[1,*], Ousmane H. Cissé[2,*,†], Jie Yun Wong[1], Peng Zheng[1], David Hewitt[3], Minou Nowrousian[4], Jason E. Stajich[2] & Gregory Jedd[1]

The advent of complex multicellularity (CM) was a pivotal event in the evolution of animals, plants and fungi. In the fungal Ascomycota, CM is based on hyphal filaments and arose in the Pezizomycotina. The genus *Neolecta* defines an enigma: phylogenetically placed in a related group containing mostly yeasts, *Neolecta* nevertheless possesses Pezizomycotina-like CM. Here we sequence the *Neolecta irregularis* genome and identify CM-associated functions by searching for genes conserved in *Neolecta* and the Pezizomycotina, which are absent or divergent in budding or fission yeasts. This group of 1,050 genes is enriched for functions related to diverse endomembrane systems and their organization. Remarkably, most show evidence for divergence in both yeasts. Using functional genomics, we identify new genes involved in fungal complexification. Together, these data show that rudimentary multi-cellularity is deeply rooted in the Ascomycota. Extensive parallel gene divergence during simplification and constraint leading to CM suggest a deterministic process where shared modes of cellular organization select for similarly configured organelle- and transport-related machineries.

[1] Temasek Life Sciences Laboratory and Department of Biological Sciences, The National University of Singapore, Singapore 117604, Singapore. [2] Department of Plant Pathology and Microbiology, Institute for Integrative Genome Biology, University of California-Riverside, Riverside, California 92521, USA. [3] Department of Botany, Academy of Natural Sciences of Philadelphia, Philadelphia, Pennsylvania 19103, USA. [4] Lehrstuhl für Allgemeine und Molekulare Botanik, Ruhr-Universität Bochum, 44780 Bochum, Germany. * These authors contributed equally to this work. † Present address: Critical Care Medicine Department, National Institutes of Health, Bethesda, Maryland 20814, USA. Correspondence and requests for materials should be addressed to J.E.S. (email: jason.stajich@ucr.edu) or to G.J. (email: gregory@tll.org.sg).

The emergence of complex multicellularity (CM) represents a major transition in the history of life on Earth. Animals, land plants and fungi independently evolved CM. Despite fundamental differences in form, physiology and development, all CM organisms possess tissues with defined three-dimensional architecture and specialized cell types[1–3]. Studies focusing on sister groups to metazoans[4–12] and land plants[13–17] have been instrumental in defining early innovations associated with the transition to CM. These include diverse protein activities mediating cell-to-cell adhesion[4,7,9,11], transcriptional regulation[7,8,10,11,13–15], signalling pathways[4–7,16,17] and spindle orientation[12]. These important advances notwithstanding, defining the molecular basis of CM remains a major challenge for cell, developmental and evolutionary biology.

In the fungi, CM is believed to have arisen twice: in the Pezizomycotina of the Ascomycota and the Agaricomycotina of the Basidiomycota[3]. Branching hyphal filaments in which cellular compartments are interconnected by perforate septa characterize these taxa. The life cycles of these fungi generally alternate between a haploid vegetative phase when hyphae form a loose network suited for invasive and foraging growth, and a sexual phase when hyphae aggregate and cells differentiate to produce multicellular fruiting bodies. This is a complex process orchestrated by multiple transcription factors and signalling pathways[18–20], indicating that CM has overlapping requirements in animals, plants and fungi.

Cytoplasmic bridges evolved independently in all eukaryotic CM taxa, suggesting that this form of intercellular communication is indispensable for CM[21]. Consistent with this idea, septal pore-associated organelles appear to be key innovations for fungal CM[22]. Open pores permit intercellular cooperation to promote rapid invasive tip growth. However, pores are sufficiently small to be gated to isolate adjacent cells. In the Pezizomycotina, this occurs through two distinct mechanisms: peroxisome-derived Woronin bodies plug the pore in response to cell lysis[23], while cytoplasm-based disordered proteins aggregate to close the pore in the context of programmed cell death[24], aging[25] and developmental differentiation[26]. The Agaricomycotina evolved an alternative solution for pore gating. In this group, the endoplasmic reticulum-derived septal pore cap is associated with pore plugging and fruiting body development[27]. All these systems have a multigenic basis. Thus, the transition to fungal CM was accompanied by the evolution of elaborate pore-gating mechanisms[22].

Many taxa within the fungi do not conform to a simple unicellular-multicellular dichotomy. Besides the CM Pezizomycotina and Agaricomycotina, close relatives of both groups show signs of significant complexity. In the Ascomycota, the Taphrinomycotina and the Saccharomycotina harbour unicellular Saccharomyces cerevisiae (budding yeast) and Schizosaccharomyces pombe (fission yeast), respectively. However, both groups also contain species having more complex forms. For example, Candida albicans (Saccharomycotina) and Schizosaccharomyces japonicus (Taphrinomycotina) can grow as yeast or hyphae[28,29], and members of the genus Taphrina (Taphrinomycotina) form a tissue resembling the spore-forming hymenium of the Pezizomycotina[30]. The genus Neolecta[31] is by far the most enigmatic in this group: it produces hyphae with perforate septa, Woronin body-like organelles and multicellular reproductive structures closely resembling those of the Pezizomycotina[31,32]. This initially led to its miss-classification as a genus within this group[31]. However, subsequent work has firmly placed Neolecta within the Taphrinomycotina[32,33]. Neolecta may, therefore, be a key species for investigating the evolution of fungal complexity.

In the present study, we sequence the Neolecta genome and employ comparative and functional genomics to identify candidate CM-associated genes. This group is enriched for functions related to diverse endomembrane organelles and their transport. Altogether, our data suggest that multicellularity is ancestral in the Ascomycota. Extensive parallel gene divergence during simplification of budding and fission yeasts, and constraint leading to CM suggest that shared aspects of cellular organization select for similarly configured organelle- and transport-related machineries.

## Results

**Neolecta irregularis fruiting bodies.** Neolecta species have not been successfully cultured in the lab. Therefore, we initiated this study by collecting Neolecta irregularis fruiting bodies in the wild and examining them by transmission electron microscopy (Fig. 1, Supplementary Fig. 1). Hyphae within fruiting bodies have septal pores that can be occluded by electron-dense structures similar to pore-associated structures observed in the Pezizomycotina. Some of these do not appear membrane-delimited, suggesting that they are cytoplasm-derived aggregates, while others are membrane-bound and similar in appearance to Woronin bodies (Fig. 1d). These data indicate that, like Neolecta vitellina[34], Neolecta irregularis possesses pore-associated organelles resembling those found in the Pezizomycotina.

**The Neolecta irregularis genome.** To investigate the genetic basis for CM in Neolecta, we sequenced genomic DNA and RNA from N. irregularis fruiting bodies. The 199-fold-coverage assembly spans 14.5 Mb with a N50 size of 16 Kb, and 44% GC content (Supplementary Table 1). The assembly is validated using the Core Eukaryotic Genes Mapping Approach[35], and shown to have 96% completeness.

Bootstrapped maximum likelihood trees constructed from a concatenation of 110 conserved single-copy genes provide definitive support for previous studies' placing Neolecta within the Taphrinomycotina[33] (Supplementary Fig. 2). The Neolecta genome is predicted to contain 5,546 protein-coding genes, with 99.4% of these being supported by RNAseq data (Supplementary Table 1). CM fungi typically encode ∼10,000 genes, while yeast genomes generally encode between 5,000 and 6,000 genes[36] (Fig. 2a). Thus, from the perspective of coding capacity, the Neolecta genome is atypical of CM fungi. We also examined the origins of gene families to estimate the degree of genetic innovation occurring upstream of various evolutionary transitions. Nodes leading to CM fungi show substantial gene family gain. By contrast, lineages leading to Neolecta and clades harbouring yeast species (Saccharomycotina and Taphrinomycotina) generally show a modest gain in gene family number and a substantial degree of loss (Fig. 2b). Despite low overall coding capacity, Neolecta harbours a significant expansion of a fungus-specific transcription factor subfamily (Supplementary Fig. 3). Fourteen members of this transcription factor subfamily appear to have arisen from lineage-specific duplication, and could account for some aspects of Neolecta's CM.

**Phylogenetic distribution of known CM-associated functions.** Using a broad sample of fungal species, including representatives from major subphyla, we next examined the phylogenetic distribution of systems experimentally shown to be required for various aspects of fungal CM. The following paragraphs sequentially cover septal pore gating, hyphal fusion, environmental sensing and developmental patterning, and cell wall biogenesis.

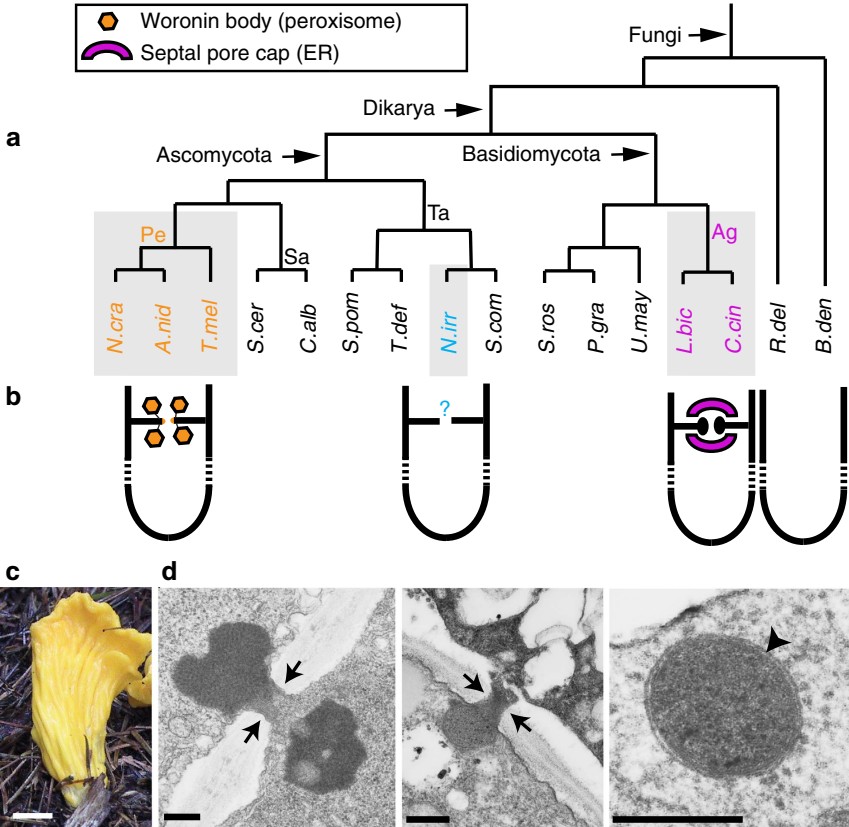

**Figure 1 | Multicellularity and septal pore gating in the fungi.** (**a**) The phylogeny of representative fungal species with sequenced genomes. CM taxa are shown with grey background. Pe, Pezizomycotina. Ag, Agaricomycotina. Sa, Saccharomycotina. Ta, Taphrinomycotina. *N. cra, Neurospora crassa. A. nid, Aspergillus nidulans. T. mel, Tuber melanosporum. S. cer, Saccharomyces cerevisiae. C. alb, Candida albicans. N. irr, Neolecta irregularis. S. com, Saitoella complicata. T. def, Taphrina deformans. S. pom, Schizosaccharomyces pombe. P. gra, Puccinia graminis. S. ros, Sporobolomyces roseus. U. may, Ustilago maydis. L. bic, Laccaria bicolor. C. cin, Coprinopsis cinerea. R. del, Rhizopus delemar. B. den, Batrachochytrium dendrobatidis.* (**b**) A simplified representation of the hypha and pore-associated membranous organelles. The colours of these organelles correspond to the text colours of the group they represent. (**c**) A *Neolecta* fruiting body collected from Black Mountain, New Hampshire. Scale bar, 5 mm. (**d**) Septal pore associated organelles of *N. irregularis*. The left panel shows an electron dense structure, which is apparently not membrane-delimited. The middle panel shows a septum that has been plugged by a Woronin body-like organelle. The right panel shows a Woronin body-like organelle free in the cytoplasm. The arrows indicate the pore and arrowheads point to the lipid bilayer. Scale bar, 250 nm. This figure is complemented by Supplementary Fig. 1, which shows additional views of the *Neolecta* fruiting body.

Genes encoding structural and regulatory proteins associated with septal-pore gating are not observed in the *Neolecta* genome. These include the Woronin body matrix protein HEX[23], its receptor WSC[37], the Leashin tether[38] and the regulator SPA-9 (ref. 24). Also absent in *Neolecta* are pore-occluding septal pore associated (SPA) proteins from the Pezizomycotina, and components of the septal pore cap of the Agaricomycotina (Fig. 3a). These data indicate that *Neolecta* is likely to have independently evolved its pore-associated organelles. This is further consistent with the presence of octahedral crystals within vacuoles (Supplementary Fig. 1d), which appear to play a role in pore gating[34].

Hyphal fusion occurs between vegetative hyphae and is associated with fruiting body development[39]. Fusion requires ancient STRIPAK[18] and MAP kinase complexes[40], which both *Neolecta* and yeasts possess (Fig. 3b). However, a group of important Pezizomycotina-specific proteins (HAM-11, HAM-5, HAM-8, SOFT, ADA-1) are absent in *Neolecta* (Fig. 3b). The SOFT protein[41], which localizes to the Woronin body in some species[42] is among these, further corroborating the likely absence of this organelle in *Neolecta*. *Neolecta* also lacks a number of other proteins (HAM-7, PRO-1, HAM-9, GAT-1, PRO-44, RCO-1) with cell fusion related functions that are ancient in the fungi.

Altogether, these data indicate that, as with septal pore gating, hyphal fusion is likely to have a distinct basis in *Neolecta*.

Environmental cues has a key role in determining developmental fate in CM fungi. White collar proteins are blue light receptors that act as transcription factors to control diverse processes including fruiting body formation[43]. Velvet family proteins are also light-regulated transcription factors that control the balance between sexual and asexual development, and coordinate development with secondary metabolism[19]. Both White collar and Velvet genes and their cofactors are mostly present in *Neolecta* and other CM fungal taxa, but absent in budding and fission yeast (Fig. 4a). This is also true for NADPH oxidases (NOX) which pattern tissues through reactive oxygen species and have been proposed to be important innovations for the evolution of multicellularity[44]. Phylogenetic trees constructed with these sequences largely reflect known evolutionary relationships, suggesting that they have been transmitted vertically during fungal evolution (Supplementary Fig. 4a–c). Altogether, these data indicate that regulation of developmental fate through these three ancient modules has been retained in *Neolecta* and CM taxa, but independently lost in the two yeast lineages. They further suggest an ancestral state of rudimentary multicellularity.

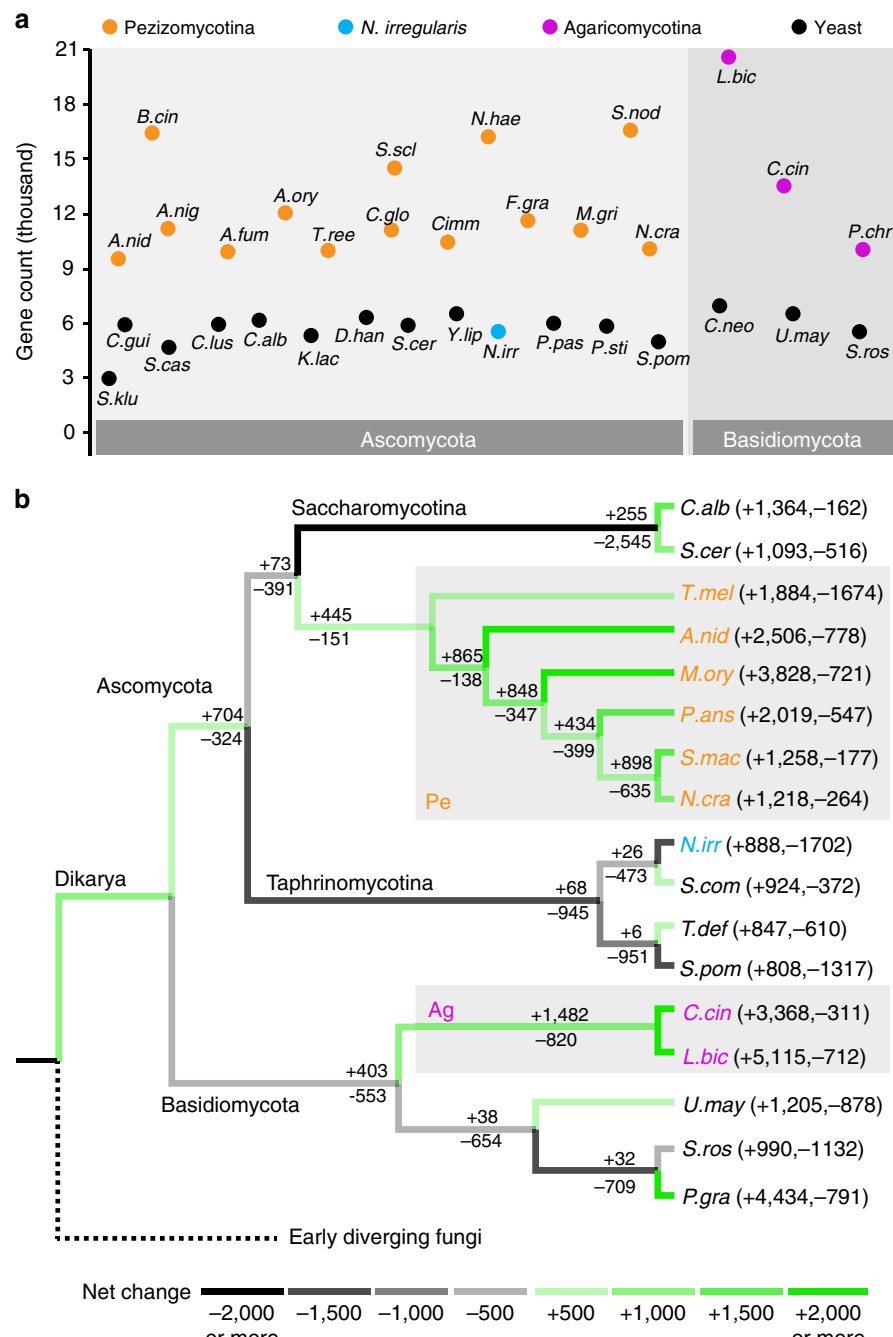

**Figure 2 | Estimated protein coding capacity of selected fungal genomes.** (**a**) The estimated number of genes is shown for selected members of the Ascomycota and Basidiomycota. *A. fum, Aspergillus fumigatus. A. nig, Aspergillus niger. A. ory, Aspergillus oryzae. B. cin, Botrytis cinerea. C. gla, Candida glabrata. C. gui, Candida guilliermondii. C. lus, Candida lusitaniae. C. glo, Chaetomium globosum. C. imm, Coccidioides immitis. C. neo, Cryptococcus neoformans. D. han, Debaryomyces hansenii. F. gra, Fusarium graminearum. K. lac, Kluyveromyces lactis. M. gri, Magnaporthe grisea. N. hae, Nectria haematococca. P. pas, Pichia pastoris. P. sti, Pichia stipitis. S. cas, Saccharomyces castellii. S. klu, Saccharomyces kluyveri. S. scl, Sclerotinia sclerotiorum. S. nod, Stagonospora nodorum. T. ree, Trichoderma reesei. Y. lip, Yarrowia lipolytica.* Other abbreviations are as indicated in the legend of Fig. 1. (**b**) Gains and losses of gene families at terminal taxa and internal nodes inferred using Dollo parsimony. Branch colour indicates the extent of net gain (green) or net loss (black) as defined in the legend.

Chitin synthases (CHS) are important for cell wall synthesis in fungi. Among the seven reported classes, CHS-5 and CHS-7 result from domain shuffling that links an extracellular CHS domain to a cytoplasmic myosin motor domain[45], providing an important innovation for hyphal morphogenesis. CHS-5 and CHS-7 are found in *Neolecta* and other CM fungal taxa, but absent in most yeasts (Fig. 4b). Again, phylogenetic trees constructed with these sequences suggest vertical transmission and independent losses from yeast genomes (Supplementary Fig. 4d). SPA-10 is another important function that regulates septum deposition[24]. It occurs exclusively in the Pezizomycotina and *Neolecta*, suggesting an origin that predates their divergence (Fig. 4b). The *Neurospora Δspa-10* mutant is defective in fruiting body development (Fig. 4c), further linking it to CM. Altogether,

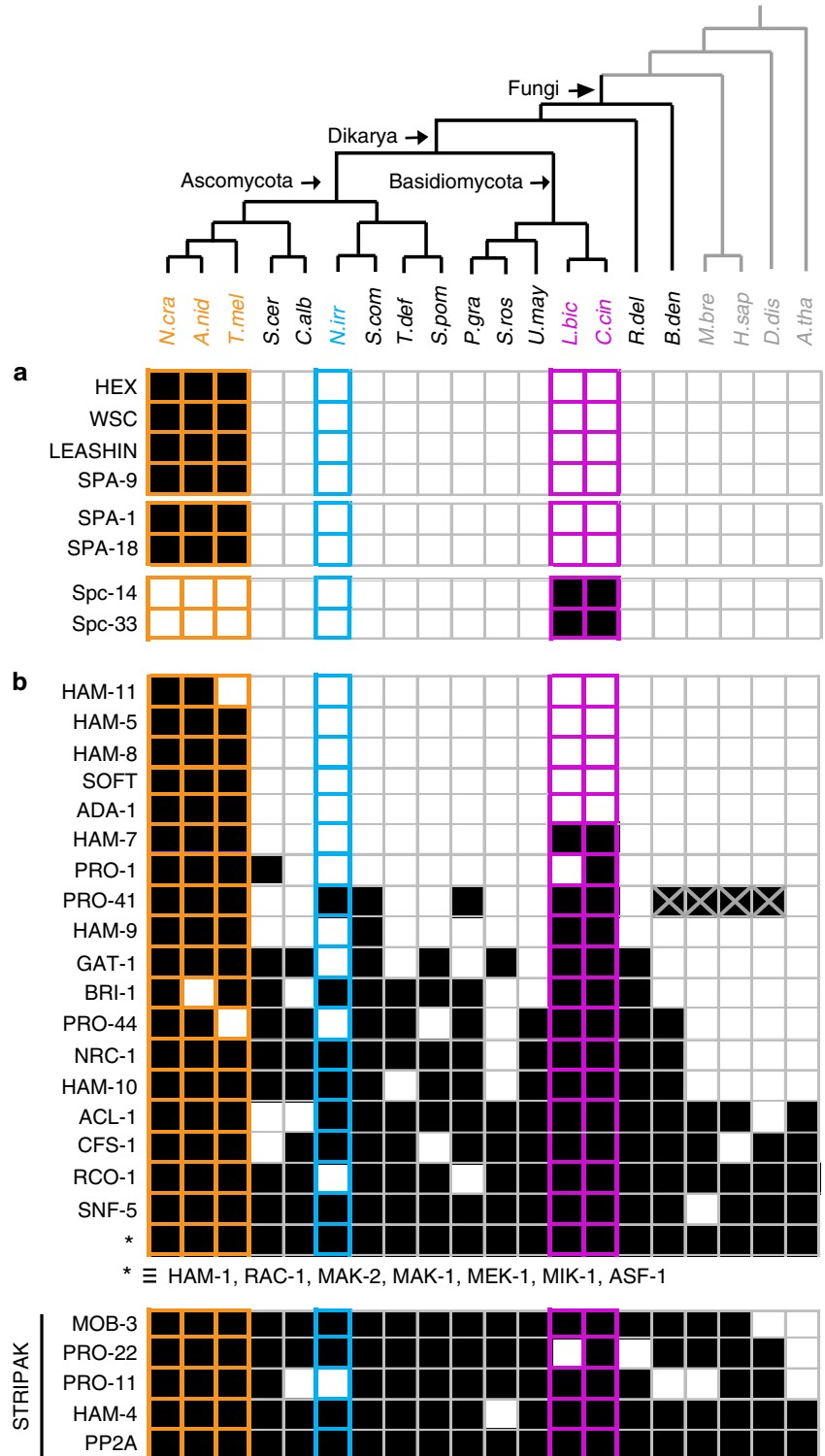

**Figure 3 | Distribution of proteins involved in septal pore gating and hyphal fusion.** (**a**) Pezizomycotina- and Agaricomycotina-specific proteins associated with septal pore gating are absent in *Neolecta*. (**b**) A subset of Pezizomycotina proteins associated with hyphal fusion (HAM-11, HAM-5, HAM-8, SOFT, ADA-1) are not found in *Neolecta*. Protein sequences are named according to the model system in which they have been most extensively characterized. For aliases, refer to Supplementary Table 6. The PRO-41 homologue in *Neolecta irregularis* was manually identified by TBLASTN. PRO41 homologues in *Batrachochytrium dendrobatidis, Homo sapiens, Monosiga brevicollis, Dictyostelium discoideum* were manually added based on published data[86]. Filled squares denote presence, empty squares denote absence. *M. bre, Monosiga brevicollis. H. sap, Homo sapiens. D. dis, Dictyostelium discoideum. A. tha, Arabidopsis thaliana.* Other abbreviations are as indicated in the legends of Figs 1 and 2.

these data suggest that *Neolecta* has retained ancestral mechanisms associated with hyphal morphogenesis and fruiting body development.

**Computational search for complexity associated proteins.** Data presented thus far indicate that *Neolecta* lacks Pezizomycotina genes associated with pore-gating and hyphal fusion. However, its

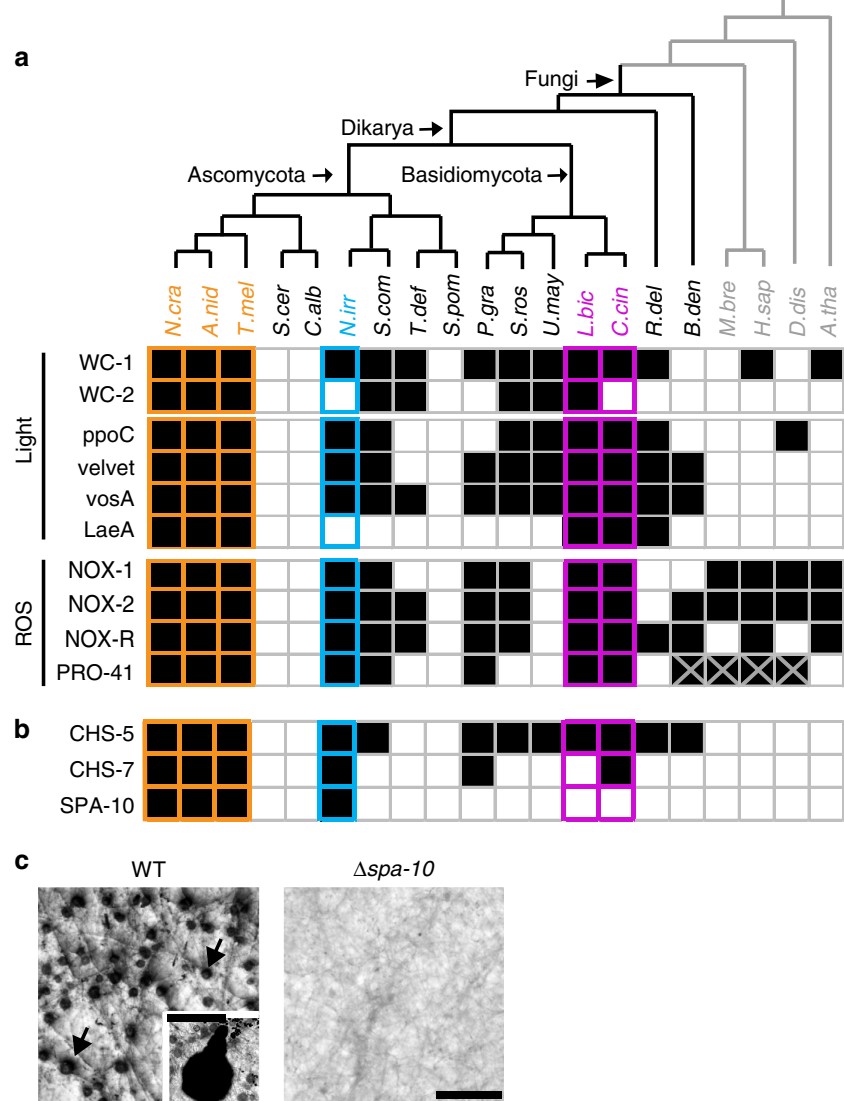

**Figure 4 | Distribution of proteins involved in signalling and hyphal morphogenesis. (a)** Genes involved in light- and ROS-related signal transduction are mostly present in *Neolecta* and other CM taxa, and absent in budding and fission yeasts. **(b)** The fusion of chitin synthase and myosin motors (CHS-5 and CHS-7) occurred early in the fungal lineage. These sequences were retained in CM taxa and lost in the two yeast lineages. The septal pore associated protein SPA-10 arose before the divergence of *Neolecta* and Pezizomycotina. Protein sequences are named according to the model system in which they have been most extensively characterized. For aliases, refer to Supplementary Table 6. Filled squares denote presence, empty squares denote absence. **(c)** Wild type (WT) hyphae make multicellular precursors (arrows), which mature into fruiting bodies upon fertilization (inset, scale bar, 100μm). The *spa-10* mutant (Δ*spa-10*) is defective in this developmental pathway. Scale bar, 1 mm.

genome encodes ancestral functions associated with environmental sensing and developmental patterning, which were lost in certain yeast lineages (Fig. 4a,b). This suggests that other CM-related genes can be discovered through their pattern of phylogenetic presence and rate of divergence. Towards this end, we used BLAST[46] and HMMER[47] to search for genes shared by the Pezizomycotina and *Neolecta*, which are either lost or show a higher degree of divergence in either budding or fission yeast (see Methods).

This search identified 1,050 genes. Remarkably, the majority of these are classified as absent (37%) or divergent (47%) in both yeast species (Supplementary Data 1). Over-represented functional categories are primarily related to endomembrane transport and organization (transport routes, substrate transport, peroxisome) and aerobic respiration (electron transport and redox-related enzymes) (Supplementary Table 2). The latter are

likely to have been lost in yeasts during their independent transitions to a facultative anaerobic lifestyle. However, because of oxygen's role in tissue patterning and secondary metabolism, a subset may also be related to CM. The genes associated with endomembrane systems and their organization include eleven dynein/dynactin motor complex[48] components (Fig. 5 and Supplementary Fig. 5), all eight subunits of the exocyst vesicle targeting complex[49], as well as other effectors of secretion and vacuolar protein sorting[50] (Supplementary Fig. 6). In the majority of cases, these are lost or more highly divergent in both yeasts.

Of the seven identified dynein-associated components, two appear to have been lost in both yeast species (p25 and p62), and five show a higher degree of divergence in both (Fig. 5a,b). One of these, p150[Glued], has a central role in regulating cargo, microtubule and dynein motor association[48]. To locate regions of sequence divergence, we devised a method for sequence

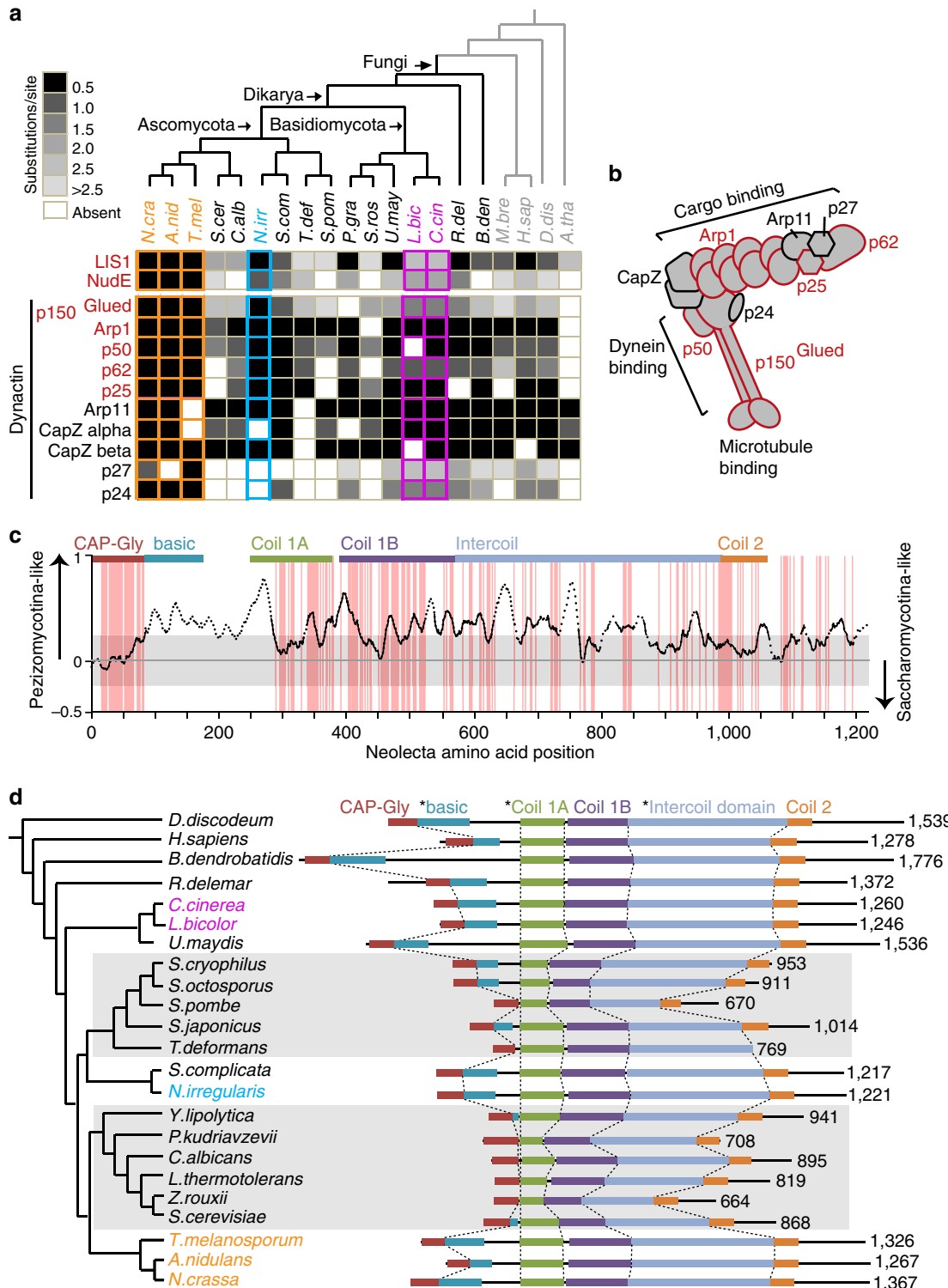

**Figure 5 | Conservation and divergence of dynein regulators.** (**a**) Substitution rate analysis of dynein regulators. For the indicated proteins, the degree of sequence divergence compared with the Pezizomycotina is determined by substitution rate and indicated by the greyscale. A lighter shade indicates greater divergence from the Pezizomycotina. Red text indicates proteins identified by our search for CM-associated sequences. This figure is complemented by Supplementary Fig. 5, which shows substitution rate analysis of dynein complex components. (**b**) The cartoon depicts the structure the Dynactin complex. Components shown in red were identified as CM-associated. (**c**) Graphical representation of the p150$^{Glued}$ multiple sequence alignment. On the vertical axis, a positive score represents greater similarity between *Neolecta* and Pezizomycotina homologues while a negative score represents greater similarity between *Neolecta* and Saccharomycotina homologues (see Methods). The grey box indicates the range of score that could be generated by chance. The dotted line represents regions missing in yeast sequences. Red background indicates residues conserved in *Neolecta* and both reference groups. p150$^{Glued}$ domains are indicated on top of the plot based on the structure of mammalian p150$^{Glued}$ (ref. 87). (**d**) Systematic length variation in p150$^{Glued}$. * indicates the regions that appear to have undergone significant contraction (Mann–Whitney *U*-test, *P*-values $<10^{-3}$) in the two yeast-containing groups (highlighted with grey background). This figure is complemented by Supplementary Fig. 7a, which shows length distribution of contracted domains.

alignment visualization (Fig. 5c). *Neolecta* p150[Glued] is more similar to Pezizomycotina orthologues over the entire length of the protein. By contrast, yeast sequences appear to have undergone domain contraction, and near complete loss of a basic domain involved in promoting motor processivity[51] (Fig. 5c,d, Supplementary Fig. 7a). Deletion of the basic domain in the model CM fungi *Neurospora crassa* (Supplementary Fig. 7b–d) and *Aspergillus nidulans*[52] results in a partial loss of function. By contrast, the reduced basic domain is dispensable for function in budding yeast[53]. Thus, budding and fission yeasts appear to have lost important domains for p150[Glued] function in the CM Ascomycota.

Our search also identified a significant number of peroxin (PEX) proteins involved in peroxisome biogenesis (Fig. 6a,b). A subset of these, PEX-33, PEX-26 and PEX-8, are not detected by similarity search in both budding and fission yeasts. However, likely functional homologues (Pex17p, Pex15p and Pex8p, respectively) have been identified in budding yeast[54]. These protein sequences have isolated regions of limited similarity to the Pezizomycotina sequences, and appear to be mostly restricted to members of the Saccharomycotina (Supplementary Fig. 8). Whether functional homologues exist in fission yeast remains unclear; nevertheless, these observations provide another striking case of overlapping genetic divergence in the two yeasts. Our search identified an additional nine peroxins that are detected in both yeasts. However, substitution rate analysis indicates greater divergence from the Pezizomycotina orthologues as compared with *Neolecta* (Fig. 6a). PEX-19 has a central role in peroxisome membrane biogenesis and was, therefore, examined in detail. PEX-19 binds peroxisome membrane proteins in the cytoplasm and delivers them to the peroxisome through an interaction with PEX-3 (ref. 55) (Fig. 6b). The PEX-3 binding segment (α-a) is highly conserved. However, sequence divergence is apparent in an amphipathic helix (α-1) required for substrate binding[56,57] (Fig. 6c,d). To determine the functional significance of this divergence, we replaced *Neurospora* PEX-19 α-1 with the corresponding region from *S. cerevisiae* (Fig. 6d). This variant displays a partial loss-of-function as indicated by fewer peroxisomes and defects in matrix protein import (Fig. 6e,f). Interestingly, the PEX-19 chaperone function has been documented in metazoans[58] and *Neurospora*[57]. However, in budding yeast, Pex19p has been associated with production of pre-peroxisomal vesicles at the endoplasmic reticulum[59]. Thus, despite a high degree of overall sequence conservation, PEX-19 appears to be functionally divergent in yeast as compared to CM-Ascomycota.

**Functional characterization of CM-associated genes**. To identify new CM-associated functions, we focused on a subset of genes identified by our search (See Methods for selection criteria, Supplementary Fig. 9a, Supplementary Data 1). Deletion strains of 147 genes were systematically examined in *Neurospora*. Seven of these have clear defects in hyphal growth and were selected for further investigation (Fig. 7a,b). Green fluorescent protein (GFP) tags at endogenous loci show that these proteins localize to distinct punctate compartments (Fig. 7c). This is consistent with the enrichment of predicted transmembrane domains (TMDs) within the overall group (Supplementary Fig. 9b) as well as these seven proteins (Fig. 7d). Four have potential homologues in animals and plants: ROGDI (NCU08091) encodes a human protein of unknown function whose mutation results in epilepsy and dementia[60], Hyphal endomembrane protein-3 (HEP-3, NCU03590) is related to Alpha/gamma-adaptin-binding protein p34 of metazoans[61], suggesting a role in clathrin-mediated protein sorting, and Vezatin (NCU09240) is involved in focal adhesions in animals[62] and has recently been shown to have a

role in hyphal polarity[63]. Finally, Mitochondria-1 (MIT-1, NCU02937) encodes a novel protein localized to mitochondria. The remaining genes encode novel fungus-specific proteins: HEP-1 (NCU03589) and HEP-2 (NCU06509) contain predicted TMDs and localize to distinct cytoplasmic puncta, and Spitzenkörper-1 (SPZ-1, NCU02049) is found at the hyphal tip where it co-localizes with markers of the Spitzenkörper vesicle supply center associated with hyphal growth and morphogenesis[64].

**Discussion**

The results presented here resolve *Neolecta*'s enigmatic relationship to the Pezizomycotina. The absence of important Pezizomycotina-specific gene families associated with pore gating and hyphal fusion indicates that *Neolecta* independently derived these functions (Fig. 3). By contrast, key genes controlling hyphal morphogenesis, environmental sensing and developmental patterning are present in *Neolecta* and the Pezizomycotina, but not in budding and fission yeasts (Fig. 4). Phylogenetic analyses support vertical transmission of these gene families (Supplementary Fig. 4). Altogether, these findings suggest that rudimentary CM is deeply rooted in the Ascomycota, and that budding and fission yeasts, as members of distinct monophyletic clades, are likely to be independently derived from a multicellular ancestor.

As a multicellular genus outside the CM Pezizomycotina (Fig. 1), *Neolecta* provides a key species for comparative genomics. By searching for gene families present in *Neolecta* and the Pezizomycotina, but absent or highly divergent in yeasts, we identified candidate genes associated with multicellularity (Supplementary Data 1). This group of 1,050 genes is enriched for functions related to diverse endomembrane organelles and their transport. These include a host of peroxisome-, dynein/dynactin- and secretion-associated functions, including all eight exocyst components (Figs 5 and 6, Supplementary Figs 5 and 6). We speculate that shared aspects of morphogenesis and development are two likely constraints on the evolution of these functions in the Pezizomycotina and *Neolecta*, while a transition to simplified cellular organization in budding and fission yeasts selected for extensive parallel gene loss and divergence. Remarkably, we also see evidence for convergence at the level of molecular structure. This is exemplified by the retention of p150[Glued] domains required for dynein motor processivity in CM fungi, and their loss or contraction in both yeasts (Fig. 5c,d). In the latter case, a diminished requirement for motor processivity is potentially related to the transition to smaller cell size. The link between peroxisomes and multicellularity (Fig. 6a) is consistent with their recognized role in Pezizomycotina sexual development[65], as well as demonstrated functions in intercellular signalling[66] and innate immunity[67] in other CM taxa. How convergent loss of peroxins alters peroxisome function in yeasts remains unclear. However, all these genes act in matrix protein import, suggesting that their loss/divergence may relate to diminished diversity of imported proteins. This is supported by the ∼2-fold lower percentage of proteins bearing the peroxisomal targeting signal 1 (PTS1) in Ascomycota yeasts compared with the CM Pezizomycotina (Supplementary Fig. 10).

Our computational search allowed for genes to be lost in either budding or fission yeast. However, the overwhelming majority of recovered genes were lost or divergent in both (84%) (Supplementary Data 1). This finding suggests a profound degree of coding capacity convergence in these two yeast clades. This is likely to be the product of shared ecological, physiological and cellular selective pressures. Convergent simplification in the these yeasts is supported by previous work demonstrating convergent loss of Complex I of the mitochondrial respiratory chain[68], and

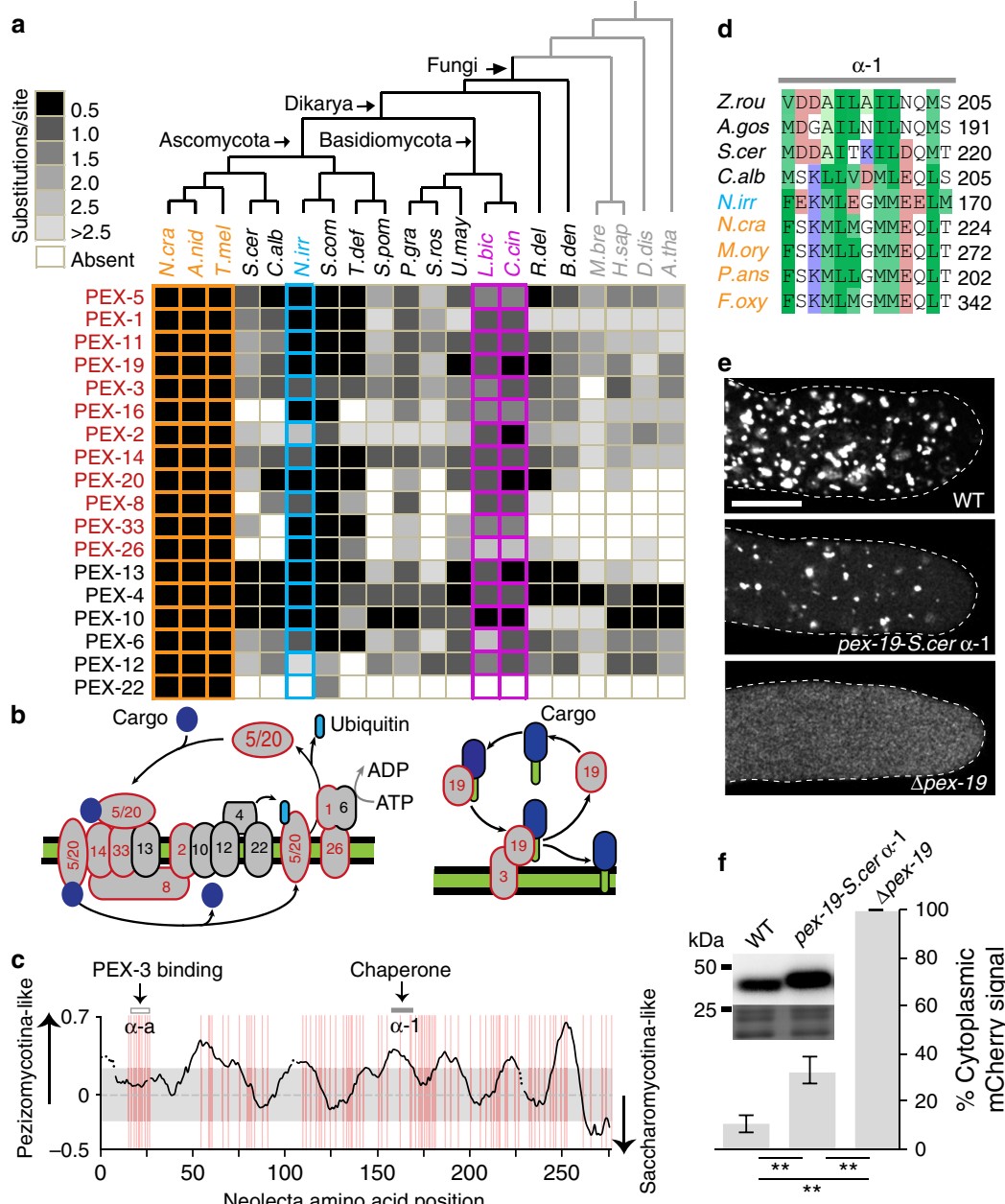

**Figure 6 | Conservation and divergence in peroxins.** (**a**) Substitution rate analysis. For the indicated peroxins, the degree of sequence divergence compared with the Pezizomycotina is determined by substitution rate and indicated by the greyscale. A lighter shade indicates greater divergence from the Pezizomycotina. (**b**) The cartoon depicts the peroxisomal matrix (left panel) and membrane protein (right panel) import machinery. Red outline indicates proteins identified by the search for CM-associated sequences. The green colour denotes the lipid bilayer and hydrophobic transmembrane domain of PEX-19 substrates. (**c**) Graphical representation of the PEX-19 multiple sequence alignment. Amphipathic segments mediating PEX-3 binding (α-a) and chaperone activity (α-1) are indicated. (**d**) Alignment of α-1 segment in members of the Saccharomycotina (black), *N. irregularis* (blue), and members of the Pezizomycotina (orange). Numbers indicate the position of the last amino acid in the corresponding sequences. Red and blue background indicates residues with negatively and positively charged side chains, respectively. Different shades of green background indicate residue hydrophobicity[88]. (**e**) Defective peroxisome biogenesis in a *Neurospora* strain expressing PEX-19 with the α-1 segment from *S. cerevisiae*. The images show the matrix marker mCherry-PTS1 in the indicated strains. Scale bar, 10 μm. (**f**) Mean level of cytoplasmic mCherry-PTS1 in the indicated strains. Error bars, s.d. (*n* = 10). Statistical significance of the difference in cytoplasmic mCherry-PTS1 levels is assessed by one-tailed *t*-test. ** indicates *P* value <10⁻³. The inset shows steady-state levels of PEX-19 as determined by western blotting (upper panel). The lower panel of the inset shows coomassie-stained bands, which serve as a loading control.

the parallel diversification of an ancestral transcription factor family[69]. Convergent phenotypic evolution with a similar genetic basis has been observed in a number of taxa. However, documented examples typically involve a limited number of genetic loci (reviewed in ref. 70). Our results show that effectors of endomembrane organization can be subject to massive convergence. This further implies that evolution can be predictable, even when targets of selection are complex and controlled by a large number of genetic loci. This finding has potentially far-reaching implications. Determining the underlying causes and generality of such predictability will require more work.

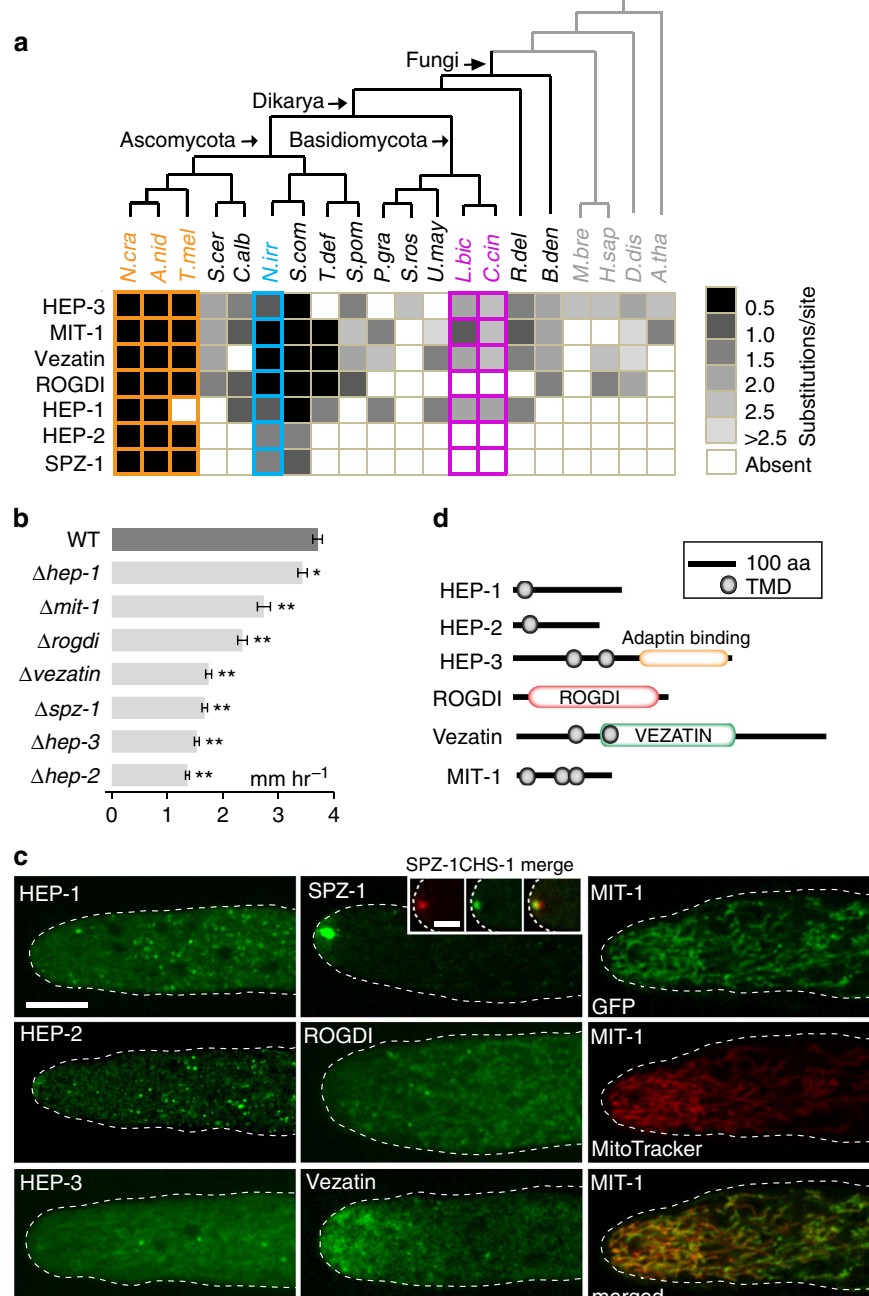

**Figure 7 | Novel CM-associated proteins important for hyphal development.** (**a**) Substitution rate analysis of novel CM-associated genes whose deletion mutants show growth defect. (**b**) Mean growth rate of wild-type and the indicated deletion strains. Error bars, s.d. ($n = 5$). The significance of growth rate difference between the wild-type and mutants is assessed using one-tailed $t$-test. * indicates $P$ value $<10^{-2}$, ** indicates $P$ value $<10^{-3}$. (**c**) The localization of the indicated proteins shown by GFP fusion at their native chromosomal loci. The dotted line indicates cell outline. Scale bar, 10 μm. The inset shows the co-localization (merge) of SPZ-1 with the Spitzenkörper marker CHS-1. Inset scale bar, 5 μm. Images in the third column show MIT-1 co-localization with the mitochondrial marker, MitoTracker. (**d**) Position of predicted transmembrane domains and known domains in the indicated proteins.

Haploid genetic model CM fungi such as *Neurospora* allow the combined use of comparative and functional genomics. This allowed us to identify previously uncharacterized genes that represent gains-of-function accompanying the transition to CM. Most of their gene products appear to be organelle-associated (Fig. 7). These include SPZ-1 (Spitzenkörper vesicle supply center), MIT-1 (mitochondria), HEP-3 (endosomes), HEP-1 and HEP-2 (uncharacterized compartments). Overall, these genes appear to have arisen throughout the evolutionary history of the lineage, predating the animal-fungi divergence, the radiation of the fungi and the *Neolecta*-Pezizomycotina divergence (Figs 4

and 7). Many of the protein functions underlying complex-ification in metazoans[4–9,11,12] and land plants[13–17,71] are present in their less complex relatives. Thus, in animals, plants and fungi, important CM-associated systems began to accumulate before the radiation of extant CM taxa.

Examination of extant taxa in animal and plant lineages generally suggests that progress towards increasingly complex forms is associated with the emergence of novel genes and gene family expansion[4–17,71]. However, because close relatives to CM groups can independently simplify[11], inferring the nature of a common ancestor is challenging. The *Neolecta* genome points to

a common multicellular ancestor in the Ascomycota. This is further consistent with the presence of many CM-associated genes in *Neolecta*'s close relative *Taphrina*, which displays significant developmental complexity[30]. *Saitoella* also possesses CM-associated genes. However, unlike *Neolecta* and *Taphrina*, *Saitoella* has only been reported to be a unicellular budding yeast[33]. This apparent incongruity could indicate that it is currently undergoing genome reduction. Alternatively, because *Saitoella* has received little attention, it may possess as-yet unrecognized complexity. In keeping with this scenario, we find that *Saitoella* can produce hypha-like cells under conditions of nitrogen starvation (Supplementary Fig. 11). This provides one basis for *Saitoella*'s gene content similarity to the CM groups. However, more work is required to understand *Saitoella*'s developmental complexity and function of its CM-associated genes.

Our computational search was designed to cast a wide net for complexity-related genes and its output is not meant to definitively identify genes associated with CM. As with Saitoella, other species outside of CM groups can possess genes identified by the search (for example, *Candida albicans*). Interestingly, the choanoflagellate and filasterean sister groups to animals do not meet the criteria of CM, yet possess many genes associated with metazoan CM[7,11]. The fungi present a spectrum of biological complexity whose genetic basis remains poorly understood, and many factors can contribute to the apparent presence or absence of a given gene. Thus, case-by-case experimental evidence is required to determine the function of a given gene in a given species.

Increasing endomembrane complexity appears to be a prerequisite for the emergence of organismal complexity. This is consistent with findings presented here (Fig. 7), as well as previous work showing that expansion of SNARE protein gene families accompanied the emergence of CM in green plants and metazoans[72,73]. Complex cellular architecture, diversified modes of cell–cell communication, and increased cell type diversity are all likely to be tied to this expansion. In the Pezizomycotina, the Spitzenkörper vesicle supply center contains concentric layers populated by distinct micro- and macro-vesicles[64,74]. This type of vesicle organization is not observed in hyphae of early diverging fungi[74]. The novel SPZ-1 protein can begin to account for this complexification. Cell polarity and secretory proteins shown to have a role in organizing the Pezizomycotina Spitzenkörper are all ancient[64], suggesting that these are unlikely to fully account for this intricate organization. SPZ-1 executes an important function (Fig. 7b) and appears to have arisen before the divergence of *Neolecta* and the Pezizomycotina (Fig. 7a). Thus, SPZ-1 can begin to account for the gains-of-function required for Spitzenkörper complexification.

Understanding the emergence of biological complexity requires the identification of relevant genes combined with knowledge of the mechanistic role played by their protein products. The former task is commonly achieved by searching for systematic gene/domain gain and loss. Our approach complements this method by also allowing for the identification of sequence variation embedded in otherwise conserved genes. Mechanistic function is often inferred from sequence similarity to known domains. As demonstrated here, combining unbiased comparative genomics with functional characterization afforded by haploid genetics allows efficient identification of complexity-associated molecular machineries. This approach also has the potential to identify the mechanisms that fostered them.

## Methods

**Phylogenetic distribution of known developmental genes.** Detailed methods on genome sequencing, assembly, annotation and phylogenetic analyses (including

gene family gain-loss and expansion analyses and identification of dynein complex component orthologues) are described in the Supplementary Methods.

The presence of homologues to known developmental genes from other fungi was detected by BLAST[46] and HMMER3 (ref. 47) (v3.1b1). HMM profiles were constructed from alignments of orthologues to characterized developmental genes in up to 22 fungal species representing Ascomycetes, Basidiomycetes, Mucoromycotina, Chytridiomycetes and Blastocladiomycetes. The HMM profiles were then used to search for homologues in the predicted proteomes of *Neolecta* and other fungi with the hmmscan function of HMMER3 ($e$-value = 1e$^{-5}$). Best hits were used for a BLAST search against the predicted *Sordaria macrospora* proteins ($e$-value = 1e$^{-5}$), and orthologues were considered as present if the protein encoded by the corresponding developmental gene from *S. macrospora* was the best hit in the reciprocal BLAST analysis.

**Identification of candidate CM-associated genes.** Protein products of genes common to the Pezizomycotina and Neolecta were identified using BLASTP[46] ($e$-value = 1e$^{-3}$) with *Neurospora* sequences as queries. Sequences were considered if they were present in at least 7 out of 14 Pezizomycotina species included. HMM profiles of these sequences were constructed using Pezizomycotina homologues using HMMER3 (ref. 47), which were then used to search for homologues in 41 representative fungal, metazoan and plant proteomes (Supplementary Table 3). Sequences were selected for further analysis if they fit these criteria: $-\log_{10}$ ($e$-value) of the best hit in *S. cerevisiae* and *S. pombe* must be lower than that of the best *N. irregularis* hit by at least 60, 40 or 20, when the best *N. irregularis* hit has $-\log 10 (e\text{-value})$ in the range (200, $+\infty$), [100,200], or [1,100), respectively (examples are shown in Supplementary Methods). Genes were selected for functional characterization if they encode a protein that is (1) present in *N. irregularis* and the Pezizomycotina, and (2) not detected in *S. cerevisiae*, *S. pombe* and non-fungal species using BLASTP (NCBI nr database, $e$-value = 1e$^{-3}$).

**Computational analyses of candidate CM-associated genes.** Functional enrichment analysis was performed using FungiFun[75]. TMDs in the *Neurospora* proteome and the set of candidate CM-associated proteins were predicted using TMpred[76]. The presence of PTS1 was predicted by searching for the signature C-terminal tripeptide [SAC][KRH]L[77] in proteins of target proteomes (Supplementary Table 4). For substitution rate analysis, a multiple sequence alignment was built for each protein and its homologue using MUSCLE[78]. The alignment was then trimmed using trimAl (ref. 79) (-gt 0.8 -cons 0.5) and used to construct a maximum likelihood tree using PhyML[80] with optimized tree topology, branch length and substitution rate. For each species $x$ in the tree, the evolutionary distance between its protein sequence and the Pezizomycotina ancestral sequence was estimated by the score, $s(x) = d(x) - \frac{\sum_{y \in Pezizomycotina} d(y)}{p}$ (unit: substitutions/site), where $d(x)$ denotes the branch length from species $x$ to the Pezizomycotina root, $p$ represents the number of Pezizomycotina species present in the tree. In words, $s(x)$ was calculated as the difference between two measures: (1) the distance from species $x$ to the Pezizomycotina root, and (2) the average distance from all extant Pezizomycotina sequences to the Pezizomycotina root. This score was used to determine the greyscale shown in Figs 5–7.

**Graphical representation of multiple sequence alignment.** Plots in Figs 5c and 6c were generated based on alignments of Pezizomycotina, *N. irregularis*, and Saccharomycotina homologues. For each position in the alignment, sum-of-pairs score of *Neolecta*-Pezizomycotina sequences ($x$) and sum-of-pairs score of *Neolecta*-Saccharomycotina sequences ($y$) were calculated following the BLOSUM62 substitution matrix[81] and a gap penalty of 3. The Pezizomycotina-similarity score for that position was the difference ($x - y$). A positive score indicates greater *Neolecta*-Pezizomycotina similarity. The scores were averaged by the Savitzky-Golay filter[82] (window = 41, 5th order polynomial), and plotted over the length of the *Neolecta* sequence. The insignificant range (grey region in the plot) was estimated by calculating the scores in the same way, for 1,000,000 randomly generated alignment columns, taking into account naturally occurring amino acid frequencies found in the BLOSUM62 matrix. A score was considered significant if it was higher than 95% of the scores resulting from those randomly generated.

**Neurospora genetics.** All *Neurospora* strains used in this study were backcrossed to wild-type strains FGSC465 or FGSC466 to obtain homokaryon mutants. Deletion mutants were obtained from the Fungal Genetics Stock Center's arrayed mutant collection[83]. CM-associated genes with growth-defective deletion mutants were GFP-tagged in FGSC9719 or FGSC9720 background using Marker Fusion Tagging[84]. To express the *p150^Glued* variant without the linker region, fusion polymerase chain reaction (PCR) was performed to join a DNA fragment encoding an HA-epitope tag and hygromycin B resistance protein to genomic DNA fragments flanking the linker region. The fusion PCR product was used to transform *Neurospora* strain FGSC9720. To express PEX-19 with *S. cerevisiae* α-1 and α-a, the region encoding these in *Neurospora pex-19* genomic DNA were replaced with those from *S. cerevisiae*. The constructs were targeted to the *his-3* locus[85] using the pBM60 vector in a Δ*pex-19* background expressing

mCherry-PTS1 (ref. 57). An HA-epitope tag was incorporated in both the variant and wild-type PEX-19 to detect their expression by western blot, using horseradish peroxidise-conjugated rat monoclonal anti-HA antibodies (Roche 12013819001, 1:2,500 dilution). Uncropped images of gels and blots, which were acquired using ChemiDoc Touch Imaging System, are shown in Supplementary Fig. 12. Strains, plasmids, and primers used to generate them are listed in Supplementary Table 5. For growth rate determination, conidia from strains of interest were inoculated on plates with Vogel's N medium. After one day of growth, agar blocks of equal dimensions were cut 2 mm behind the colony's growth front and placed inside race tubes fashioned from 25 ml disposable pipettes containing the same medium. The average growth rate per hour was calculated from the distance travelled by the colony's growth front after 48 h.

**Image acquisition and analysis.** Images shown in Figs 6e and 7c were taken using a Leica SP8 inverted confocal microscope with the HCX PL APO 100 × /1.40 OIL objective and acquired by Leica Application Suite X. To measure the level of cytoplasmic mCherry-PTS1 signal and construct the graph shown in Fig. 6f, images of 10 growing hyphal tips randomly selected from each strain were obtained using fixed microscope settings. ImageJ (http://rsb.info.nih.gov/ij/) was used to quantify the total and peroxisomal mCherry-PTS1 signals. The difference between these gives the cytoplasmic signal.

**Code availability.** Computer scripts used to generate and process data presented in this paper can be found at https://github.com/ocisse/Neolecta_genome_project.

**Data availability.** The *Neolecta irregularis* whole-genome shotgun project has been deposited at DDBJ/ENA/GenBank under the accession LXFE00000000. The assembly can be reached through NCBI (https://www.ncbi.nlm.nih.gov/) using the accession no. LXFE01000000, BioProjectID 167926. The species tree shown in Supplementary Fig. 2 and the concatenated alignment used to generate it have been deposited at TreeBase (http://purl.org/phylo/treebase/phylows/study/TB2:S19388). Other data relevant to the findings presented here are available upon request.

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

## Acknowledgements

Research in the Jedd group is funded by the Temasek Life Sciences Laboratory and Singapore Millennium Foundation. J.E.S. was supported by the A.P. Sloan Foundation, USDA Agriculture Experimental Station at the University of California-Riverside and NIFA Hatch project CA-R-PPA-5062-H. M.N. was supported by the German Research Foundation (DFG NO407/5-1) and thanks Ulrich Kück for his support at the Department of General and Molecular Botany. O.H.C was supported by the Swiss National Science Foundation fellowship grant no. 151780. We thank Zhang Louxin for insightful discussion, Dillon McDonald and Katherine Borkovich for help with *Saitoella* experiments, Don Pfister, Pete and Kitty Griffith for help with *Neolecta* fruiting bodies collection, and gratefully acknowledge use of deletion mutants generated by National Institutes of Health Grant P01 GM068087 'Functional Analysis of a Model Filamentous Fungus'.

## Author contributions

G.J. conceived the project. G.J. and D.H. collected *Neolecta irregularis* fruiting bodies. G.J. and J.E.S. obtained *Neolecta* sequence. O.H.C. and J.E.S. performed genome assembly, annotation, fungal phylogeny construction and gene family gain/loss/expansion analysis. O.H.C., J.E.S. and M.N. examined the phylogenetic distribution of known CM-associated proteins. T.A.N. performed computational searches for CM-associated functions, graphical representation of multiple sequence alignments and substitution rate analysis. T.A.N., J.Y.W. and P.Z. employed functional genomics to characterize novel CM-associated proteins. T.A.N. performed genetic manipulations of *pex19* and *p150^Glued*. O.H.C. and J.E.S. performed experiment on *Saitoella complicata*. G.J. wrote the manuscript with input from all coauthors.

## Additional information

**Competing financial interests:** The authors declare no competing financial interests.

