## [Peer Review File · Nature Communications]

Reviewers' comments:

Reviewer #1 (Remarks to the Author):

This interesting work describes the sequencing and characterization of the *Neoelecta irregularis* genome a species phylogenetically related to early diverging yeasts. This fungal species has unique features with respect to functions related to complex multi-cellularity (CM). In particular, it shares features of CM with fungi in a distinct lineage (Pezizomycotina) that produce multicellular fruiting bodies in addition to an independent evolutionary route to a septal pore gating mechanism. They leverage these features in conjunction with its relatedness to budding and fission to understand the evolutionary forces and mechanisms driving the emergence of CM and the simplification leading to the unicellular yeast form. In addition to the striking example of convergent evolution of septal pore gating in the *Neoelecta* lineage. Their comparative and functional approach including the phenotypic characterization has resulted in novel and important insights into the functions involved in these processes and the evolutionary forces that shaped them.

The manuscript is clearly written and the figures easily interpreted. The methodology used to do both the sequence analysis and the subsequent phylogenetic analysis are state-of-the-art. I can't honestly find one substantive weakness in this work and think it should be accepted in its current form.

Reviewer #2 (Remarks to the Author):

The paper by Jedd et al addresses the genetic bases of fungal multicellularity. The model system chosen (*Neoelecta*) seems to be an enigmatic multicellular lineage among fungi that offers a cool system to ask questions like what genes are needed for multicellular growth. However, I found a number of fundamental issues with the approach used and with the interpretation of the results.

First of all, I think it would be helpful if the authors clarified somewhere in the ms how they define complex multicellularity. I repeatedly noticed that the authors's definition of CM differ from Knoll's widely accepted definition (2011 *Ann. Rev. E. Plan. Sci*), which made it difficult at times to follow the author's conclusions. Further, the authors seem to switch back and forth between alternative definitions. On page 6 they look for genes related to fruiting body development, which is complex multicellularity sensu Knoll, whereas elsewhere in the ms they look for genes involved in hyphal morphogenesis, which is different from how complex multicellularity is usually defined. In my opinion hyphal growth falls closer to simple multicellularity, although its a matter of definitions. In any case, the authors should focus on either fruiting body development or hyphal growth and clarify which is the focus of their study.

I also found the analyses problematic. The authors look for genes shared by *Neoelecta* and *Pezizomycotina* but lost in unicellular yeasts (though not all, see below). This approach

assumes that the ancestor of Neolecta and the Peizomycotina was also complex multicellular. I found a paper by Healy et al (Mycologia. 2013 Jul-Aug;105(4):802-13. doi: 10.3852/12-347) that shows that complex multicellularity in Neolecta is the result of convergent evolution. If so, then why do the authors expect to find shared genes that are lost in yeasts? Rather, convergent/parallel gene innovations should be sought for.

Another problem with the analyses is that for flagging a gene as CM-related, the pipeline requires gene losses in only 2 out of 4 yeasts species and presence in only 1 out of several complex multicellular species present in the dataset. The authors seem to not consider the fact that many of the 1000 genes they identify as CM-related are present and conserved in yeasts (*Sporobolomyces* and *Saitoella*, see *comments below) and also many are absent or divergent in complex multicellular Basidiomycota (*Laccaria* and *Coprinopsis*, but also *Tuber*). Although they marked these taxa as CM on their figures, no requirement for CM-related genes to be present is built into the pipeline. Why was that?

I can imagine this liberal approach to finding CM-related genes being the reason why only 4% of the predicted genes (7 out of 147) showed an actual growth defect in their forward genetic study. However, this calls into question the reliability of the predictions.

Nevertheless, the observation that Neolecta has evolved complex multicellularity independently is extremely interesting. Having evolved CM with a basically yeast-like, very compact genome is also very interesting. By showing that known CM-related genes are absent in Neolecta, the authors have convincingly demonstrated that Neolecta evolved CM separately, which, however, has been known in the literature. I found the presentation of the genome of Neolecta being the most solid part of the ms, maybe that could be expanded in greater detail?

I also found several imprecise phrasings, like 'broad sample', 'substantial gene loss', etc. throughout the manuscript.

Specific comments:

p3l16 - even earlier diverging clades (e.g. *Mucoromycotina*) produce hyphae with perforate septa, so this is not something that makes Neolecta enigmatic

p3l23 - please provide a reference for the independent origins of unicellular growth in yeasts. Was it truly independent simplification, or budding and fission yeasts being 2 plesiomorphically simple lineages from which the more complex fungi evolved?

p3l24 - in this paragraph the authors draw a parallel between cell-cell channels of plants and animals and septal pores of fungi. To me this is somewhat problematic since septal pores develop to compartmentalize the hyphal lumen with an additional benefit of allowing cell-cell communication, as opposed to cell-cell channels which develop to enable cell-cell communication. Please reconsider this comparison

p3l26 - can you back this statement up by a reference?

p4l10 - fungi that do not produce fruiting bodies also have pore-gating mechanisms, which suggests to me that pore-gating is not a requirement to CM per se.

p4l15 - 'lineage'

p4|21 - the paragraph 'Cytology of *Neolecta irregularis* fruiting bodies' only focuses on the structure of septal pores. A more appropriate paragraph title is needed. I feel the authors could have expanded more on the structure and function of these septal occlusions. Also, Supplementary Fig 1 shows asci, an image of hyphae in the core of the fruiting body and an octahedral crystal in a vacuole. How do these relate to the biology of *Neolecta*? Please explain.

p5|4 - A previous study (Healy et al Mycologia) concluded that the pore associated organelles of *Neolecta* are analogous to those found in the Pezizomycetes. Thus, 'resembling' might be misleading here. Please rephrase, or explain if the conclusion is different from the Healy et al paper.

p5|8 - in the supplementary material Supp. Table 1 comes after S. table 5. There is a large difference between the number of RNA-Seq based transcripts and the number of predicted genes. What could be causing this?

p5|18 and throughout the ms - italicize '*neolecta*' and other generic names

p5|20 - the authors mention gene gain here, whereas in the methods gene family gain and loss is mentioned (p16|6). Please clarify. Would it be possible to more phrase more specifically instead of just saying 'substantial'?

p5|26 - this is speculation

p6|2 - what do the authors mean by 'broad sample'?

p6|17 - the term Pezizomycotina-specific implies that they are missing in other lineages. The same on Fig 3., if its Agaricomycotina-specific then its absence in *Neolecta* is not surprising. Also on Fig 3, what Pezizomycotina gene does *bri1* refer to? The only reference I could find for *bri1* is a transcription factor of Basidiomycota (*Schizophyllum*, see Ohm 2011 Mol. Microb.). If so, its not Pezizomycotina-specific (as the Fig caption states)

p6|14 - this paragraph is not entirely clear to me. First, l15-16 states that the STRIPAK and MAP kinase complexes are present in *Neolecta*, yet, the authors conclude on l22 that hyphal fusion has a 'distinct basis' in *Neolecta*. Is this not a contradiction? Second, the paragraph head mentions fruiting body development, yet, this is not expanded on in the paragraph. Further, the sentence on SOFT protein could move to the previous paragraph, since its Woronin-body associated.

p6|20 - what are these 'other' genes? Are they different from the ones listed on Fig 3?

*p7|3 - well, based on Fig4a they are absent in one yeast lineage, but present in *Schizosaccharomyces*, *Neolecta*'s immediate phylogenetic neighbor.

*p7|4 - The same applies to NOX-es, they are present in *Schizosaccharomyces*. Further, both NOX-es and WC and velvet proteins are present in *Sporobolomyces*, a third yeast lineage in the Basidiomycota (according to Fig 3a).

*p7|10 - again, I can't see how the authors see these being lost in 'two yeast lineages'? They are present in *Schizosaccharomyces* - in fact, *Schizosaccharomyces* has more of these proteins than *Neolecta*. A third yeast lineage (*Sporobolomyces roseus*, which was missed by the authors) also has a complete set of these proteins. The presence of these genes in yeasts suggests that they do not necessarily confer CM-specific functions. I suggest eliminating this paragraph.

*p7|15 - well, hyphal morphogenesis per se is not complex multicellularity, since simple multicellular fungi also possess hyphal morphogenesis genes. Again, out of 3 yeast lineages only 1 has lost CHS-5 and CHS-7 (*Schizosaccharomyces* and *Sporobolomyces* have it acc. to fig4), which suggests that these are not CM-specific and cannot be used as an

argument pro *Neolecta*'s retained ancestral cell wall biogenesis mechanisms.

Why is PRO41 present on both Fig3 and Fig4?

p8l2 - please delete the second part of the sentence.

*p8l6 - it is somewhat surprising that the authors restricted their analyses to genes lost in *Saccharomyces* and *Schizosaccharomyces*, while their dataset included three other yeasts, *Candida albicans*, *Saitoella complicata* and *Sporobolomyces roseus* also. If their hypothesis about the loss of CM-related genes holds, why not require the absence of CM-related genes in these species too in the analyses? Further, looking for genes shared by *Neolecta* and the *Pezizomycotina* implies a single origin of CM, which is in conflict with the general view that *Neolecta* evolved CM independently (Healy et al *Mycologia* 2013) and also with the authors' results that suggest independent gene gains in CM fungi (p5l20). If a single origin of CM is assumed, then why not include CM Basidiomycota in the analyses too?

p8l7 please provide exact figures. Also, I don't think this should be particularly 'remarkable' given that the search criteria required these to be absent or divergent in yeasts.

*Looking at Supplementary Table 2, a number of observations can be made. First, of the 1054 genes detected, most are present in *Saitoella complicata* or *Taphrina deformans* or both (columns M&N). *Saitoella* is a yeast and *Taphrina* is dimorphic, neither of them produces fruiting bodies, yet they possess these genes. If the detected genes are indeed CM-related why are they present in these yeast species? Another observation that can be made is that many of the 1054 genes have no copies in typical CM species like *Tuber melanosporum* (column I), *Coprinopsis cinerea* and *Laccaria bicolor* (columns S&T). The search criteria in their current form seem to be too permissive (e.g. homology only between *Neurospora* and *Neolecta* is required), I wonder if more specific results could be obtained if the authors required that the candidate CM genes should be present in more than 1 examined CM species? For example, a gene to be designated as candidate CM gene, its presence in CM basidiomycota, CM Ascomycota but its absence in all yeasts and early-diverging fungi could be required. This would obviously result in a much shorter gene list, but probably with a much lower false positive rate.

p8l14-19 and suppl. Fig 5 - The listed complexes/mechanisms sound more like components of basic cellular processes (mRNA transport, spindle orientation ,etc..) than anything specific to complex multicellularity. On Fig S5 why are lines 1-6 in this table marked as CM associated? I can't see how substrate transport or nutrient perception would be related to complex multicellularity.

*p8l18 and Fig S7 - how do the authors explain the fact that CM basidiomycete copies of these genes are at least as divergent from *Pezizomycete* copies as are yeast proteins? Again, the yeast *Saitoella* does not show divergence in these proteins, which contradicts their role in CM (as does their divergence in CM basidiomycetes).

*p8l22 - but they are present in other yeasts, such as *Candida albicans*, *Saitoella* or *Sporobolomyces*

p10l7-12 - this concluding section should reflect on the conservation of many of these proteins in other yeast species (e.g. *Saitoella*, *Sporobolomyces*) and the divergence of these in CM Basidiomycetes, as exceptions from the rule and that these potentially put a question

mark on the role of these gene families in CM.

p10|13 - I am confused by the use of the term complex multicellularity by the authors. I thought CM refers to species forming 3-dimensional structures, in accordance with Knoll 2011, but they mention hyphal CM here.

p10|17 - does the deletion of the other 140 genes not result in a visible phenotype?

p10|18 - again, a working definition of complex multicellularity would be useful. If above the authors looked for genes involved in fruiting body development, why do they consider defects in hyphal growth here?

p10|20 I CAN'T SEE WHY THEY SHOW MEMBRANE ASSOCIATED LOCALIZATION??????

p11|12 morphogenesis and developmental patterning might be too strong statements relative to what genes were detected. Could the authors more specifically state what genes were detected? In general, I strongly question whether the analyzed gene families are related to CM. Maybe fungal-specific traits would be a better phrase instead of CM, since a clear link to complex multicellularity is missing and most genes were either present in some yeasts and/or missing from basidiomycete CM taxa.

p11|21 - again, what is CM and what is hyphal CM?

p12|2 - again links CM to sexual development (fruiting bodies). This is confusing. The authors need to clarify whether they look for genes involved in hyphal multicellularity or fruiting body production. In its current form, the ms jumps back and forth between hyphal multicellularity and fruiting body production, providing a mixture of gene families related to either one.

p12|5 - the authors should mention here what percentage of these genes were NOT lost in *Candida* or *Saitoella* and what percentage was lost or divergent in basidiomycete CM taxa. Please acknowledge where the pattern does not fit CM.

p12|17 - the relationship to cell size is speculative, please use more conservative phrasing

p12|18-20 - how is simplification consistent with the expansion of a TF family?

p12|24 - well, throughout the manuscript the authors were talking about coincidental losses in yeasts (and that's what the analytical pipeline is geared towards). If lost in yeasts, it must have been present in the common ancestor of *Neolecta* and *Pezizomycotina*, in which case it's not convergence.

p12|24 - I don't think these results go as far as implying the predictability of evolution. That hypothesis should be tested on different systems.

p13|12 - where was SNARE expansion shown in this ms?

p13|24 - this would be a very interesting aspect of the paper, so please show the data in some form.

p14|1-10 - Can the authors demonstrate that *Saitoella* indeed diverged recently from *Neolecta* (e.g. by molecular clock analyses)? Assuming that *Saitoella* evolved the yeast-like growth recently would imply that all other yeast-like species of the *Taphrinomycotina* had *Neolecta*-like CM ancestors. This is an unlikely assumption since 99% of the *Taphrinomycotina* is yeast-like.

p14|11 - it could, but what evidence other than copy-numbers support this idea?

p14|21-23 - could the authors provide a reference for this proposition?

p15|10 - please specify the method/kits used. How were libraries prepared?

p15|13 - please provide details of the RNA-Seq

p15|22 - how were these 110 orthologs selected?

p17113-14 - how were these parameters chosen?

Please note that I do not enumerate all the things that need to be corrected in the methods section. I find the methods' description a bit superficial.

Reviewer #3 (Remarks to the Author):

NCOMMS-16-17154-T

REVIEW: Innovation and constraint leading to complex multicellularity in the Ascomycota.

AUTHORS: T Nguyen, O Cissé, J Wong, P Zheng, D Hewitt, M Nowrousian, J Stajich, G Jedd

BACKGROUND: The advent of multicellularity was a major evolutionary transition in the history of life, opening the door to cellular differentiation and division of labor, and promoting a dramatic increase in biocomplexity. Unlike mitochondriogenesis, a transition that appears to have succeeded only once, multicellularity has arisen at least 25 times, and in every major eukaryotic lineage. While the body plans of plants, animals and fungi exhibiting complex multicellularity (CM) have been elaborated in very different ways, at the cellular and subcellular levels the major lineages share many features, highlighting both the antiquity and utility of this major transition.

MODEL & APPROACH: The fact that multicellularity had so many independent origins ideally suits comparative genomics to the task of answering longstanding questions about the composition of the toolbox that made CM possible, and whether different tools can be selected for, and still lead clades to converge on the fungal model for CM. Especially useful in this undertaking are enigmatic species that lie well off the beaten phylogenetic path, exhibit CM, but have odd features that make them "neither fish nor fowl." All the better if the sister taxa to such organisms have well-annotated genomes, and are amenable to standard genetic manipulations such as gene deletion or gene knockdown. Such a taxon is the fungal genus *Neolecta*, which exhibits CM, but whose genome size and coding capacity is half that of other CM fungi. In the fungal kingdom CM manifests as hyphal cells interconnected via septal pores+reproduction via multicellular fruiting bodies. In the Ascomycota, CM emerged in the Pezizomycotina, represented by *Neurospora*, *Aspergillus* and *Tuber*. Despite being phylogenetically affiliated with early diverging yeasts, *Neolecta* possesses Pezizomycotina-like CM. Nguyen et al. sequence the *Neolecta irregularis* genome and identify ancient CM-associated functions by searching for genes that are conserved in *Neolecta* and the Pezizomycotina, but absent or divergent in budding and fission yeasts.

FINDINGS: The approach of comparative genomics carried out in conjunction with *Neurospora* functional analyses uncovered interesting patterns of presence/absence among CM-related genes related to hyphal development, and also revealed extensive conservation and divergence in dynein regulators, peroxisome (Woronin body) functioning. The fact that Basidiomycota, *Neolecta* and Ascomycota such as *Neurospora* share multiple genes in hyphal morphogenesis, as well light- and ROS-dependent signaling indicates that CM is deeply rooted in the Ascomycota, and secondarily lost in species such like *C. albicans* and *S. cerevisiae*. This interpretation is supported by findings that a number of genes needed in *Neurospora* for septal pore gating, hyphal fusion and sexual development are absent from

Neoelecta and budding yeast (as well as many Basidiomycota).

IMPRESSIONS: I can count on one hand the number of times I have used the word "masterful" to describe a MS I review for the first time. This is a masterful piece of science: Nguyen et al. pose a big question, then answer it thoroughly using complementary phylogenetic analyses and functional studies. The manuscript is well written, and the figures attractive and compelling. At first I was taken aback by the sheer volume of data and was concerned that this would be a "steamroller contribution" in which the authors sought to overwhelm reviewers with so much information that the reviewers would ultimately have to yield. But after some effort I recognized that this was a concise offering, making just a few important points. Though I am favorably disposed towards this MS, I ask that the authors to respond to the following criticisms.

1. Of late, there has been considerable buzz in the evolutionary biology community about experimental evolution of multicellularity in Bakers yeast. This work has been extolled by some, but criticized by others, usually on the point that *S. cerevisiae* likely had a multicellular ancestor. Because there are data in Nguyen et al. that directly speak to this controversy, they should do so, especially if they aspire to publication in a generalist journal like Nature Commun.

2. I object to the naïve use of "evolutionary success in the following statement: "lineage leading to the Pezizomycotina shows evidence for significant accumulation of new gene families, and has radiated to produce tens of thousands of species with great morphological, ecological and lifestyle diversity. By contrast, Neoelecta comprises only three highly related species. The comparison between these two clades supports the proposition that genetic innovation and evolutionary success are positively correlated." This statement suggests higher-order selection, tantamount to saying the Lemuridae, which encompasses many species are more evolutionarily successful than the Hominidae, which has but few. I would argue a taxon's "evolutionary success" resides more in its persistence than in its proclivity to diversify.

3. I am uncomfortable with these statements (p. 10, ll. 7-12): "In summary, these data suggest that cellular architecture can profoundly influence the evolutionary fate of peroxisome-, dynein/dynactin- and secretion-associated functions (Figs. 5 and 6, Supplementary Fig. 8a, Supplementary Table 3). Hyphal organization of the Pezizomycotina and Neoelecta is likely to have constrained the evolution of these functions, while a transition to simplified cellular organization in budding and fission yeasts selected for extensive parallel gene loss and divergence." And the repetitive: "These data suggest that hyphal CM profoundly constrained the evolution of organelles and transport functions. Shared cellular architecture and the need for long distance transport are two aspects that are likely to provide this constraint." While I am sympathetic to this view, the authors have a bit of a "chicken or egg" problem (genes first or structure first). I can imagine profound evolutionary consequences following the loss or cooption one or few genes at the point when these clades diverged – and indeed such loss or cooption could have driven their initial divergence.

4. Nguyen et al. amply cite the work of King and others working on the evolution of animal multicellularity. They should more clearly and creatively situate their findings in relation to those studies and studies on CM in plants. What have they achieved that has not been achieved in those systems? What general patterns have they uncovered that contrast or complement the findings of studies in the other eukaryotic kingdoms?

5. Finally, the authors pass up the opportunities to speculate in their discussion as to where next their approach could be fruitfully applied (and why) – and to comment on new directions they would like to pursue or see others pursue.

We thank the reviewers for their thoughtful feedback and taking the time to help us improve our manuscript. Our responses are found highlighted in grey below (References are included in the form of PubMed ID).

Reviewers' comments:

Reviewer #1 (Remarks to the Author):

This interesting work describes the sequencing and characterization of the *Neoelecta irregularis* genome a species phylogenetically related to early diverging yeasts. This fungal species has unique features with respect to functions related to complex multicellularity (CM). In particular, it shares features of CM with fungi in a distinct lineage (Pezizomycotina) that produce multicellular fruiting bodies in addition to an independent evolutionary route to a septal pore gating mechanism. They leverage these features in conjunction with its relatedness to budding and fission to understand the evolutionary forces and mechanisms driving the emergence of CM and the simplification leading to the unicellular yeast form. In addition to the striking example of convergent evolution of septal pore gating in the *Neoelecta* lineage. Their comparative and functional approach including the phenotypic characterization has resulted in novel and important insights into the functions involved in these processes and the evolutionary forces that shaped them.

The manuscript is clearly written and the figures easily interpreted. The methodology used to do both the sequence analysis and the subsequent phylogenetic analysis are state-of-the art. I can't honestly find one substantive weakness in this work and think it should be accepted in its current form.

Reviewer #2 (Remarks to the Author):

The paper by Jedd et al addresses the genetic bases of fungal multicellularity. The model system chosen (*Neoelecta*) seems to be an enigmatic multicellular lineage among fungi that offers a cool system to ask questions like what genes are needed for multicellular growth. However, I found a number of fundamental issues with the approach used and with the interpretation of the results.

First of all, I think it would be helpful if the authors clarified somewhere in the ms how they define complex multicellularity. I repeatedly noticed that the authors's definition of CM differ from Knoll's widely accepted definition (2011 *Ann. Rev. E. Plan. Sci*), which made it difficult at times to follow the author's conclusions. Further, the authors seem to switch back and forth between alternative definitions. On page 6 they look for genes related to fruiting body development, which is complex multicellularity sensu Knoll, whereas elsewhere in the ms they look for genes involved in hyphal morphogenesis, which is different from how complex multicellularity is usually defined. In my opinion hyphal growth falls closer to simple multicellularity, although its a matter of definitions. In any case, the authors should focus on either fruiting body development or hyphal growth and clarify which is the focus of their study.

I also found the analyses problematic. The authors look for genes shared by Neoelecta and Pezizomycotina but lost in unicellular yeasts (though not all, see below). This approach assumes that the ancestor of Neoelecta and the Peizomycotina was also complex multicellular. I found a paper by Healy et al (Mycologia. 2013 Jul-Aug;105(4):802-13. doi: 10.3852/12-347) that shows that complex multicellularity in Neoelecta is the result of convergent evolution. If so, then why do the authors expect to find shared genes that are lost in yeasts? Rather, convergent/parallel gene innovations should be sought for.

Another problem with the analyses is that for flagging a gene as CM-related, the pipeline requires gene losses in only 2 out of 4 yeasts species and presence in only 1 out of several complex multicellular species present in the dataset. The authors seem to not consider the fact that many of the 1000 genes they identify as CM-related are present and conserved in yeasts (*Sporobolomyces* and *Saitoella*, see *comments below) and also many are absent or divergent in complex multicellular Basidiomycota (*Laccaria* and *Coprinopsis*, but also *Tuber*). Although they marked these taxa as CM on their figures, no requirement for CM-related genes to be present is built into the pipeline. Why was that?

I can imagine this liberal approach to finding CM-related genes being the reason why only 4% of the predicted genes (7 out of 147) showed an actual growth defect in their forward genetic study. However, this calls into question the reliability of the predictions.

Nevertheless, the observation that Neoelecta has evolved complex multicellularity independently is extremely interesting. Having evolved CM with a basically yeast-like, very compact genome is also very interesting. By showing that known CM-related genes are absent in Neoelecta, the authors have convincingly demonstrated that Neoelecta evolved CM separately, which, however, has been known in the literature. I found the presentation of the genome of Neoelecta being the most solid part of the ms, maybe that could be expanded in greater detail?

I also found several imprecise phrasings, like 'broad sample', 'substantial gene loss', etc. throughout the manuscript.

All of the comments found above are repeated in the specific comments below and are responded to there.

Specific comments:

p3116 - even earlier diverging clades (e.g. *Mucoromycotina*) produce hyphae with perforate septa, so this is not something that makes Neoelecta enigmatic

We disagree with this perspective: The case for *Neoelecta*'s being enigmatic is not based solely on its perforate septa, rather, on the combination of its placement in the Taphrinomycotina, and its possession of multiple features characteristic of the CM Pezizomycotina. Please see the revised last paragraph of introduction for further information (page 4, lines 6-20).

p3123 - please provide a reference for the independent origins of unicellular growth in yeasts. Was it truly independent simplification, or budding and fission yeasts being 2 plesiomorphically simple lineages from which the more complex fungi evolved?

We agree: It was an oversight to bring this point up at the level of the introduction. Independent simplification of these two yeasts is now examined in the discussion of the revised manuscript.

p3l24 - in this paragraph the authors draw a parallel between cell-cell channels of plants and animals and septal pores of fungi. To me this is somewhat problematic since septal pores develop to compartmentalize the hyphal lumen with an additional benefit of allowing cell-cell communication, as opposed to cell-cell channels which develop to enable cell-cell communication. Please reconsider this comparison

We disagree with this perspective: Firstly, septa, rather than septal pores, compartmentalize the hyphal lumen. Secondly, we make this point to highlight the exploitation of cell-cell channels in the major CM taxa. That they do not entirely overlap in their multiple functions is not surprising and irrelevant to the point we are making.

p3l26 - can you back this statement up by a reference?

This statement is substantiated by the set of referenced sentences that follow it.

p4l10 - fungi that do not produce fruiting bodies also have pore-gating mechanisms, which suggests to me that pore-gating is not a requirement to CM per se.

The statement is correct within the context of this paragraph. We are not claiming that pore-gating mechanisms alone are sufficient for CM, which seems to be suggested by this comment.

p4l15 - 'linage'

The typo has been corrected.

p4l21 - the paragraph 'Cytology of *Neolecta irregularis* fruiting bodies' only focuses on the structure of septal pores. A more appropriate paragraph title is needed. I feel the authors could have expanded more on the structure and function of these septal occlusions. Also, Supplementary Fig 1 shows asci, an image of hyphae in the core of the fruiting body and an octahedral crystal in a vacuole. How do these relate to the biology of *Neolecta*? Please explain.

The term "cytology" is not essential to this heading and has been removed. The image of asci is included in the supplement to give readers unfamiliar with fungal fruiting bodies a better idea of their structure. For a discussion of the biology of the octahedral crystal please see page 6, lines 12-15.

p5l4 - A previous study (Healy et al *Mycologia*) concluded that the pore associated organelles of *Neolecta* are analogous to those found in the Pezizomycetes. Thus, 'resembling' might be misleading here. Please rephrase, or explain if the conclusion is different from the Healy et al paper.

Healy *et. al.*, examines pore-associated structures solely based on microscopy. Without molecular evidence, neither homology nor analogy can be inferred. The pore-associated organelles we observe closely resemble Woronin bodies of the Pezizomycotina. Our use of the word 'resembling' is conservative, claiming only what is evident in the data.

p5l8 - in the supplementary material Supp. Table 1 comes after S. table 5. There is a large difference between the number of RNA-Seq based transcripts and the number of predicted genes. What could be causing this?

Response to the first comment: Following *Nature Communications* submission guidelines, Supplementary Table 1 is short and therefore, included in a file together with other supplementary materials. Other Supplementary Tables are much longer and are therefore submitted as a separate Excel file.

Response to the second comment: In *de novo* assembly, it is common that several transcripts/isoforms correspond to a single predicted gene. The difference between the numbers of transcripts and genes is not unusual.

p5l18 and throughout the ms - italicize 'neolecta' and other generic names

This has been done.

p5l20 - the authors mention gene gain here, whereas in the methods gene family gain and loss is mentioned (p16l6). Please clarify. Would it be possible to more phrase more specifically instead of just saying 'substantial'?

Response to the first comment: We examined gene family gains and losses and have edited is section to harmonize the terminology. Please see page 5, lines 22-26 and page 16, line 26.

Response to the second comment: The text summarizes what is presented in the figure. The word 'substantial' describes the number of gene family gains at nodes leading to extant CM, relative to nodes leading to extant yeasts. The accompanying figure is meant to provide the exact numbers.

p5l26 - this is speculation

This comment refers to our finding of an expanded transcription factor family in *Neolecta* and our suggestion that this expansion could account for some aspects of *Neolecta*'s CM. While we agree that this is speculative, it is a reasonable hypothesis supported by our data.

p6l2 - what to the authors mean by 'broad sample'?

We have clarified the meaning. Please see Page 6, lines 5-6.

p6l17 - the term Pezizomycotina-specific implies that they are missing in other lineages. The same on Fig 3., if its Agaricomycotina-specific then its absence in *Neolecta* is not surprising. Also on Fig 3, what Pezizomycotina gene does bri1 refer to? The only reference I could find for bri1 is a transcription factor of Basidiomycota (*Schizophyllum*,

see Ohm 2011 Mol. Microb.). If so, its not Pezizomycotina-specific (as the Fig caption states)

In this passage, we refer to a group of Pezizomycotina-specific genes, which are the first 5 genes in Fig. 3a. In the legend of Fig. 3, we only claim that a subset of genes are Pezizomycotina-specific. *bri1* does not belong to this subset. To avoid such confusion, we have edited the text to list these Pezizomycotina-specific sequences. Please see page 6, lines 19-20.

p6l14 - this paragraph is not entirely clear to me. First, l15-16 states that the STRIPAK and MAP kinase complexes are present in Neoelecta, yet, the authors conclude on l22 that hyphal fusion has a 'distinct basis' in Neoelecta. Is this not a contradiction? Second, the paragraph head mentions fruiting body development, yet, this is not expanded on in the paragraph. Further, the sentence on SOFT protein could move to the previous paragraph, since its Woronin-body associated.

We disagree with this perspective: First point - Hyphal fusion requires many genes of differing phylogenetic age. Our data show that Neoelecta possesses ancient hyphal fusion associated genes (STRIPAK and MAP Kinases), but lacks a number of important Pezizomycotina-specific genes. The latter finding supports the suggestion that hyphal fusion in Neoelecta has a distinct basis.

Second point - It is correct that SOFT is physically associated with the Woronin body, but the significance of this association remains unclear. *Soft* loss-of-function results in female sterility and an inability to produce fruiting bodies. Thus, its inclusion in the analysis of fruiting body-associated genes is appropriate.

p6l20 - what are these 'other' genes? Are they different from the ones listed on Fig 3?

'Other' genes refer to genes that are not Pezizomycotina-specific. The text has been changed to clarify this point. Please see page 6, line 23.

*p7l3 - well, based on Fig4a they are absent in one yeast lineage, but present in Schizosaccharomyces, Neoelecta's immediate phylogenetic neighbor.

*p7l4 - The same applies to NOX-es, they are present in Schizosaccharomyces. Further, both NOX-es and WC and velvet proteins are present in Sporobolomyces, a third yeast lineage in the Basidiomycota (according to Fig 3a).

*p7l10 - again, I can't see how the authors see these being lost in 'two yeast lineages'? They are present in Schizosaccharomyces - in fact, Schizosaccharomyces has more of these proteins than Neoelecta. A third yeast lineage (*Sporobolomyces roseus*, which was missed by the authors) also has a complete set of these proteins. The presence of these genes in yeasts suggests that they do not necessarily confer CM-specific functions. I suggest eliminating this paragraph.

These comments state that certain genes (*nox-1*, *nox-2*, *pro-41*, *white-collar-1*, *white collar-2*) are present in *Schizosaccharomyces pombe*. This is incorrect. Data presented in Figure 4 indicate absence. With respect to *Sporobolomyces roseus*, CM Basidiomycetes possess highly differentiated cells involved in aerial spore discharge. *Sporobolomyces roseus* also possesses the ability to differentiate these cells. Thus, we do not consider it exemplary of a unicellular yeast.

*p7115 - well, hyphal morphogenesis per se is not complex multicellularity, since simple multicellular fungi also possess hyphal morphogenesis genes. Again, out of 3 yeasts lineages only 1 has lost CHS-5 and CHS-7 (*Schizosaccharomyces* and *Sporobolomyces* have it acc. to fig4), which suggests that these are not CM-specific and cannot be used as an argument pro *Neolecta*'s retained ancestral cell wall biogenesis mechanisms.

The comment is incorrect. The data in Figure 4 indicate absence of these genes in *Schizosaccharomyces pombe*. As per *Sporobolomyces roseus*, please refer to the last two sentences of our response to the preceding comment.

Why is PRO41 present on both Fig3 and Fig4?

PRO41 has functions that fit the general categories in both Figs. 3 and 4. Its inclusion is therefore appropriate.

p812 - please delete the second part of the sentence.

Since this request is made without any context, it is unclear what is being objected to. The second part of the sentence is supported by our data, which show that these genes are indeed lost in the two yeast lineages (Figs. 3 and 4).

*p816 - it is somewhat surprising that the authors restricted their analyses to genes lost in *Saccharomyces* and *Schizosaccharomyces*, while their dataset included three other yeasts, *Candida albicans*, *Saitoella complicata* and *Sporobolomyces roseus* also. If their hypothesis about the loss of CM-related genes holds, why not require the absence of CM-related genes in these species too in the analyses? Further, looking for genes shared by *Neolecta* and the Pezizomycotina implies a single origin of CM, which is in conflict with the general view that *Neolecta* evolved CM independently (Healy et al Mycologia 2013) and also with the authors' results that suggest independent gene gains in CM fungi (p5120). If a single origin of CM is assumed, then why not include CM Basidiomycota in the analyses too?

Candida is not "a yeast" as stated in this comment, rather it is dimorphic and can grow as yeast or hyphae. We have shown that *Saitoella* can produce hypha-like cells (Supplementary Fig. 10) and *Sporobolomyces* possesses features of related CM Basidiomycetes (see above). For these reasons we do not consider these species as exemplary of unicellular yeast.

We strongly disagree with the statement that there is a "general view that *Neolecta* evolved CM independently (Healy et al Mycologia 2013)". Healy *et. al.* exclusively used electron microscopy to examine *Neolecta vitellina* septal pore-associated structures. Without an association between genes and these morphological features, this idea is unsupported.

p817 please provide exact figures. Also, I don't think this should be particularly 'remarkable' given that the search criteria required these to be absent or divergent in yeasts.

The manuscript clearly states that the search allowed the gene to be absent or divergent in *either* yeast. The comment implies that the search required the genes to be absent or divergent in *both* yeasts. We are also not clear on why a precise number is requested here. The exact figures can be found from Supplementary Table 3.

*Looking at Supplementary Table 2, a number of observations can be made. First, of the 1054 genes detected, most are present in *Saitoella complicata* or *Taphrina deformans* or both (columns M&N). *Saitoella* is a yeast and *Taphrina* is dimorphic, neither of them produces fruiting bodies, yet they possess these genes. If the detected genes are indeed CM-related why are they present in these yeast species? Another observation that can be made is that many of the 1054 genes have no copies in typical CM species like *Tuber melanosporum* (column I), *Coprinopsis cinerea* and *Laccaria bicolor* (columns S&T). The search criteria in their current form seem to be too permissive (e.g. homology only between *Neurospora* and *Neolecta* is required), I wonder if more specific results could be obtained if the authors required that the candidate CM genes should be present in more than 1 examined CM species? For example, a gene to be designated as candidate CM gene, its presence in CM basidiomycota, CM Ascomycota but its absence in all yeasts and early-diverging fungi could be required. This would obviously result in a much shorter gene list, but probably with a much lower false positive rate.

1. This comment seems to be based on the view that results generated by computational searches should be definitive answers to the question 'What constitutes CM?'. We view computational genomics as a means of hypothesis generation, not as an end in itself.

2. The comment points out that CM-associated genes are found in *Saitoella* and *Taphrina*. *Saitoella* has not been extensively studied, and we show that it produces hypha-like cells (Supplementary Fig. 10), thus its level of complexification remains unclear. After *Neolecta*, *Taphrina* is likely to be the second most complexified member of the Taphrinomycotina. *Taphrina* is not simply dimorphic as stated in the reviewer comment, rather, its hyphae also form a tissue resembling the hymenium of the CM fungi. In the revised manuscript, the issue of complexification in other early-diverging members of the Ascomycota is now introduced in the last paragraph of the introduction (page 4, lines 6-20) and further elaborated upon in a new paragraph in the discussion (page 13, lines 15-26 and page 14, lines 1-3).

Another important point regarding the presence of CM-associated genes in less complex relatives is that evolution is ongoing. Many of these species may be in the process of simplification. Thus, fully understanding the significance of any gene in a particular species requires functional characterization. We make this point on page 13, lines 21-26 and page 14, lines 1-3.

3. The comment suggests that we should have made presence in CM Basidiomycetes part of our search criteria. We strongly disagree with this suggestion. Our analysis was focused primarily on CM in the Ascomycota. Had we done as suggested, many of the most interesting genes would have been excluded from analysis (see Figure 7). Please also refer to statement 1 above.

p8114-19 and suppl. Fig 5 - The listed complexes/mechanisms sound more like components of basic cellular processes (mRNA transport, spindle orientation, etc..) than anything specific to complex multicellularity. On Fig S5 why are lines 1-6 in this table

marked as CM associated? I can't see how substrate transport or nutrient perception would be related to complex multicellularity.

We disagree with this comment: Recent work has shown how specific changes in the mechanisms that control spindle orientation are directly tied to the emergence of complex tissues in animals (PMID: 26740169). This comment fails to acknowledge this and other work supporting the view that CM emerges in association with complexification in cellular and subcellular processes. This is also a main point of our paper.

*p8l18 and Fig S7 - how do the authors explain the fact that CM basidiomycete copies of these genes are at least as divergent from Pezizomycete copies as are yeast proteins? Again, the yeast *Saitoella* does not show divergence in these proteins, which contradicts their role in CM (as does their divergence in CM basidiomycetes).

This is the result of their evolutionary relationship. The Basidiomycota-Ascomycota split is much more ancient than the *Neolecta-Saitoella* divergence, so the fact that *Neolecta*, and by extension, *Saitoella* homologs are less divergent from Pezizomycotina homologs should not be surprising. With respect to *Saitoella*, please see our discussion on page 13, lines 21-26 and page 14, lines 1-3.

*p8l22 - but they are present in other yeasts, such as *Candida albicans*, *Saitoella* or *Sporobolomyces*

We have responded to a similar point earlier. Please see our response to the comment highlighted in yellow above.

p10l7-12 - this concluding section should reflect on the conservation of many of these proteins in other yeast species (e.g. *Saitoella*, *Sporobolomyces*) and the divergence of these in CM Basidiomycetes, as exceptions from the rule and that these potentially put a question mark on the role of these gene families in CM.

Please see our response to the comment highlighted in yellow above.

p10l13 - I am confused by the use of the term complex multicellularity by the authors. I thought CM refers to species forming 3-dimensional structures, in accordance with Knoll 2011, but they mention hyphal CM here.

We have made changes throughout the text to clarify our definition of CM.

p10l17 - does the deletion of the other 140 genes not result in a visible phenotype?

The relevance of this question is unclear. The lack of a visible phenotype in a deletion mutant under a single laboratory condition does not indicate dispensability or a lack of function.

p10l18 - again, a working definition of complex multicellularity would be useful. If above the authors looked for genes involved in fruiting body development, why do they consider defects in hyphal growth here?

We have clarified our definition of complex multicellularity. In brief, we do not view this as a simple dichotomy, but rather as a spectrum of complexity (please see last paragraph of the introduction). We did not look for genes involved in fruiting body development, neither did our materials and methods contain any information that could lead to this misunderstanding.

p10I20 I CAN'T SEE WHY THEY SHOW MEMBRANE ASSOCIATED LOCALIZATION??????

The conclusion that these proteins are endomembrane-associated is based on a combination of computationally predicted transmembrane domains, punctate localization and co-localization with known markers of endomembrane compartments. These are standard procedure to demonstrate endomembrane association.

p11I12 morphogenesis and developmental patterning might be too strong statements relative to what genes were detected. Could the authors more specifically state what genes were detected? In general, I strongly question whether the analyzed gene families are related to CM. Maybe fungal-specific traits would be a better phrase instead of CM, since a clear link to complex multicellularity is missing and most genes were either present in some yeasts and/or missing from basidiomycete CM taxa.

We disagree that morphogenesis and developmental patterning are inappropriate terms to apply to the genes shown in Figure 4. NOX and its regulators have been shown to alter developmental patterning in response to oxygen cues (PMID: 18567788), White collar proteins allow light regulation of a variety of developmental transitions (PMID: 6235211, 9115195) and Velvet family proteins control the balance between sexual and asexual development (PMID: 18556559). CHS proteins and SPA-10 have previously been shown to control septum biogenesis and spatial patterning (PMID: 25596036, 22955885). Finally, figure 4C shows that SPA-10 is required for fruiting body development.

p11I21 - again, what is CM and what is hyphal CM?

We have revised the manuscript to remove the term "hyphal CM".

p12I2 - again links CM to sexual development (fruiting bodies). This is confusing. The authors need to clarify whether they look for genes involved in hyphal multicellularity or fruiting body production. In its current form, the ms jumps back and forth between hyphal multicellularity and fruiting body production, providing a mixture of gene families related to either one.

This passage cites published work linking peroxisomes to specific aspects of fruiting body development. Hyphal multicellularity provides the basis for fruiting body development. We do not agree that they should be treated as separate.

p12I5 - the authors should mention here what percentage of these genes were NOT lost in *Candida* or *Saitoella* and what percentage was lost or divergent in basidiomycete CM taxa. Please acknowledge where the pattern does not fit CM.

We have responded to a similar point about these yeasts earlier. Please see our response to “*p8l6 - it is somewhat surprising that the authors restricted their analyses to genes lost in *Saccharomyces* and *Schizosaccharomyces*” and to the comment highlighted in yellow above.

p12l17 - the relationship to cell size is speculative, please use more conservative phrasing

This speculation is based on the known role of the Dynein/Dynactin motor in long distance transport and the loss of regions associated with motor processivity in both yeasts. We have softened the statement by replacing “likely” with “potentially”. The revised sentence reads as follows: “the contraction of the basic domain promoting motor processivity⁶⁴ is potentially related to the transition to smaller cell size.”

p12l18-20 - how is simplification consistent with the expansion of a TF family?

This idea is not our own. It is derived from Nagy *et. al.* (2014), which is cited at the end of the sentence.

p12l24 - well, throughout the manuscript the authors were talking about coincidental losses in yeasts (and thats what the analytical pipeline is geared towards). If lost in yeasts, it must have been present in the common ancestor of *Neolecta* and *Pezizomycotina*, in which case its not convergence.

The convergence is not between *Neolecta* and the *Pezizomycotina*. We refer to convergent loss of genes in these two yeasts, which is supported by the data, despite the search criteria allowing loss in either *S. cerevisiae* or *S. pombe*. (Supplementary Table 3).

p12l24 - I don't think these results go as far as implying the predictability of evolution. That hypothesis should be tested on different systems.

This idea is supported by our data and presented as a hypothesis. As such, its discussion is appropriate.

p13l12 - where was SNARE expansion shown in this ms?

It is not shown, but cited in this manuscript. The statement occurs in the context of the discussion about the general tendency of endomembrane complexification associated with the emergence of CM. We have reworded the passage for greater clarity.

p13l24 - this would be a very interesting aspect of the paper, so please show the data in some form.

We disagree. Details concerning Spitzenkorper marker mislocalization are tangential to the main points of the paper.

p14l1-10 - Can the authors demonstrate that *Saitoella* indeed diverged recently from *Neolecta* (e.g. by molecular clock analyses)? Assuming that *Saitoella* evolved the yeast-

like growth recently would imply that all other yeast-like species of the Taphrinomycotina had *Neoelecta*-like CM ancestors. This is an unlikely assumption since 99% of the Taphrinomycotina is yeast-like.

The recent divergence of *Saitoella* from *Neoelecta* is strongly supported by the phylogenetic tree presented in Supplementary Fig. 2. The statement that “99% of the Taphrinomycotina are yeast-like” is incorrect. In fact, a substantial fraction of the Taphrinomycotina display complex life-cycles. This is exemplified by the hymenium forming *Taphrina deformans*, hypha forming *Schizosaccharomyces japonicus*, and *Pneumocystis*, which can differentiate ameboid-like cells.

p14l11 - it could, but what evidence other than copy-numbers support this idea?

We responded to a similar point about these transcription factors earlier. The statement is based solely on copy-number and is phrased in a conservative manner. As such, we believe it is an acceptable hypothesis.

p14l21-23 - could the authors provide a reference for this proposition?

This passage has been removed.

p15l10 - please specify the method/kit used. How were libraries prepared?

The requested information is now included. Please see Page 16, lines 6-12.

p15l13 - please provide details of the RNA-Seq

Additional information is now found in the materials and methods. Please see page 16, lines 9-11. A full accounting of the transcripts and assembly can be found in GenBank (accession: LFXE01000000) and NCBI Sequence Read Archive (accession: SRX247597).

p15l22 - how were these 110 orthologs selected?

They were selected because they are conserved single-copy orthologs. This is a standard procedure for phylogeny reconstruction (PMID: 25274300, 26580012, 18709599).

p17l13-14 - how were these parameters chosen?

The use of BLAST e-values as a measure of divergence is not unprecedented. For example, Aravind *et. al.*, 2000 (PMID: 11016957) used a difference of 10 orders of magnitude as an indication of divergence. At 20 orders of magnitude difference, our measure is more stringent.

Please note that I do not enumerate all the things that need to be corrected in the methods section. I find the methods' description a bit superficial.

We disagree. Since there is no specific comment here, we have nothing further to add.

Reviewer #3 (Remarks to the Author):

NCOMMS-16-17154-T

REVIEW: Innovation and constraint leading to complex multicellularity in the Ascomycota.

AUTHORS: T Nguyen, O Cissé, J Wong, P Zheng, D Hewitt, M Nowrousian, J Stajich, G Jedd

BACKGROUND: The advent of multicellularity was a major evolutionary transition in the history of life, opening the door to cellular differentiation and division of labor, and promoting a dramatic increase in biocomplexity. Unlike mitochondriogenesis, a transition that appears to have succeeded only once, multicellularity has arisen at least 25 times, and in every major eukaryotic lineage. While the body plans of plants, animals and fungi exhibiting complex multicellularity (CM) have been elaborated in very different ways, at the cellular and subcellular levels the major lineages share many features, highlighting both the antiquity and utility of this major transition.

MODEL & APPROACH: The fact that multicellularity had so many independent origins ideally suits comparative genomics to the task of answering longstanding questions about the composition of the toolbox that made CM possible, and whether different tools can be selected for, and still lead clades to converge on the fungal model for CM. Especially useful in this undertaking are enigmatic species that lie well off the beaten phylogenetic path, exhibit CM, but have odd features that make them “neither fish nor fowl.” All the better if the sister taxa to such organisms have well-annotated genomes, and are amenable to standard genetic manipulations such as gene deletion or gene knockdown. Such a taxon is the fungal genus *Neolecta*, which exhibits CM, but whose genome size and coding capacity is half that of other CM fungi. In the fungal kingdom CM manifests as hyphal cells interconnected via septal pores+reproduction via multicellular fruiting bodies. In the Ascomycota, CM emerged in the Pezizomycotina, represented by *Neurospora*, *Aspergillus* and *Tuber*. Despite being phylogenetically affiliated with early diverging yeasts, *Neolecta* possesses Pezizomycotina-like CM. Nguyen et al. sequence the *Neolecta irregularis* genome and identify ancient CM-associated functions by searching for genes that are conserved in *Neolecta* and the Pezizomycotina, but absent or divergent in budding and fission yeasts.

FINDINGS: The approach of comparative genomics carried out in conjunction with *Neurospora* functional analyses uncovered interesting patterns of presence/absence among CM-related genes related to hyphal development, and also revealed extensive conservation and divergence in dynein regulators, peroxisome (Woronin body) functioning. The fact that Basidiomycota, *Neolecta* and Ascomycota such as *Neurospora* share multiple genes in hyphal morphogenesis, as well light- and ROS-dependent signaling indicates that CM is deeply rooted in the Ascomycota, and secondarily lost in species such like *C. albicans* and *S. cerevisiae*. This interpretation is supported by findings that a number of genes needed in *Neurospora* for septal pore gating, hyphal fusion and sexual development are absent from *Neolecta* and budding yeast (as well as many Basidiomycota).

IMPRESSIONS: I can count on one hand the number of times I have used the word “masterful” to describe a MS I review for the first time. This is a masterful piece of

science: Nguyen et al. pose a big question, then answer it thoroughly using complementary phylogenetic analyses and functional studies. The manuscript is well written, and the figures attractive and compelling. At first I was taken aback by the sheer volume of data and was concerned that this would be a “steamroller contribution” in which the authors sought to overwhelm reviewers with so much information that the reviewers would ultimately have to yield. But after some effort I recognized that this was a concise offering, making just a few important points. Though I am favorably disposed towards this MS, I ask that the authors to respond to the following criticisms.

1. Of late, there has been considerable buzz in the evolutionary biology community about experimental evolution of multicellularity in Baker's yeast. This work has been extolled by some, but criticized by others, usually on the point that *S. cerevisiae* likely had a multicellular ancestor. Because there are data in Nguyen et al. that directly speak to this controversy, they should do so, especially if they aspire to publication in a generalist journal like Nature Commun.

We would not criticize experimental evolution of multicellularity in baker's yeast so much based on *S. cerevisiae* having a multicellular ancestor. Rather, we question whether these lab experiments can be conducted with sufficient population size and on time scales that can capture the gains-of-function that underlie much evolutionary complexification. From this perspective, we feel that the work on yeast laboratory evolution is a bit too tangential to our paper to warrant discussion. We agree that it is an interesting topic and hope that the paper stimulates commentary and discussions along these lines.

2. I object to the naïve use of “evolutionary success in the following statement: “lineage leading to the Pezizomycotina shows evidence for significant accumulation of new gene families, and has radiated to produce tens of thousands of species with great morphological, ecological and lifestyle diversity. By contrast, Neolecta comprises only three highly related species. The comparison between these two clades supports the proposition that genetic innovation and evolutionary success are positively correlated.” This statement suggests higher-order selection, tantamount to saying the Lemuridae, which encompasses many species are more evolutionarily successful than the Hominidae, which has but few. I would argue a taxon's “evolutionary success” resides more in its persistence than in its proclivity to diversify.

We agree with this perspective. Because this discussion is peripheral to our main conclusions, it has been removed from the revised manuscript.

3. I am uncomfortable with these statements (p. 10, ll. 7-12): “In summary, these data suggest that cellular architecture can profoundly influence the evolutionary fate of peroxisome-, dynein/dynactin- and secretion-associated functions (Figs. 5 and 6, Supplementary Fig. 8a, Supplementary Table 3). Hyphal organization of the Pezizomycotina and Neolecta is likely to have constrained the evolution of these functions, while a transition to simplified cellular organization in budding and fission yeasts selected for extensive parallel gene loss and divergence.” And the repetitive: “These data suggest that hyphal CM profoundly constrained the evolution of organelles and transport functions. Shared cellular architecture and the need for long distance transport are two aspects that are likely to provide this constraint.” While I am sympathetic to this view, the authors have a bit of a “chicken or egg” problem (genes first

or structure first). I can imagine profound evolutionary consequences following the loss or cooption one or few genes at the point when these clades diverged – and indeed such loss or cooption could have driven their initial divergence.

The comment refers to some of the manuscript's central findings. As such, we feel that it is important for us to discuss them. In the revised manuscript, we remove the repetition by deleting the passage from the results section. We agree that the original wording may have been too strong, and qualify our view by referring to it as speculation. With respect to the chicken or egg problem, we agree that many scenarios could be envisioned for how the phylogenomic endpoints we describe were arrived at. However, this consideration should not undermine the narrative since we are pointing out the association between shared aspects of biology and patterns of gene loss/divergence and retention. The new passage reads as follows: "We speculate that shared aspects of morphogenesis and development are two likely constraints on the evolution of these functions in the Pezizomycotina and *Neolecta*, while a transition to simplified cellular organization in budding and fission yeasts selected for extensive parallel gene loss and divergence."

4. Nguyen et al. amply cite the work of King and others working on the evolution of animal multicellularity. They should more clearly and creatively situate their findings in relation to those studies and studies on CM in plants. What have they achieved that has not been achieved in those systems? What general patterns have they uncovered that contrast or complement the findings of studies in the other eukaryotic kingdoms?

We agree with this point and have revised the manuscript in the following ways.

1. We have added a number of references to more fully cite the work on the evolution of complexity in plants. (page 3 lines 6-9, page 13 lines 12 and 17)
2. We compare our findings with both those in animals and plants to conclude that "important CM-associated systems began to *accumulate* prior to the radiation of extant CM taxa." (page 13, lines 13-14), and that "Increasing endomembrane complexity appears to be a prerequisite for the emergence of organismal complexity" (page 13, lines 4-5).
3. Because close relatives to CM groups can be independently simplified, inferring the nature of a common ancestor is challenging. We discuss this problem and insights afforded by the *Neolecta* genome in the 5th paragraph of the discussion (page 13, lines 15-26 and page 14, lines 1-3).
4. Our approach goes beyond current practice in two important ways. **A.** Unlike previous work, which typically looks at gene presence and absence, our approach allows the identification of sequence variation embedded in otherwise conserved genes. **B.** Functionally characterized domains are typically used to infer gene function. Because most of these are defined in model organisms, this approach does not account for the hidden complexity present in less-studied groups. We show that new CM-associated functions can be uncovered by combining unbiased comparative genomics with functional characterization through haploid genetics. The last paragraph of the discussion makes these two points (page 14, lines 20-26 and page 15, lines 1-3).

5. Finally, the authors pass up the opportunities to speculate in their discussion as to where next their approach could be fruitfully applied (and why) – and to comment on new directions they would like to pursue or see others pursue.

We have revised the last paragraph to emphasize how our approach allows the discovery of new multicellularity associated functions, and its potential in laying the groundwork for identifying new CM-associated mechanisms. We have also written some additional material along the lines of new directions, but found that it either sounds self-promotional or reads like a commentary on the work. As mentioned in response to comment #1, we hope that our work engenders discussion and commentary on how aspects of our approach can impact other fields and areas of investigation.

Reviewers' comments:

Reviewer #2 (Remarks to the Author):

Review of the revised ms by Nguyen et al on Neolecta. I think this is a great topic, a very interesting model and there is no doubt the ms contains several important findings. On the otherhand, I was a bit surprised by the authors' dismissal of my suggestion to include all yeasts and primarily unicellular species in their analyses. Their dataset contains 5 species that grow primarily as unicellular and are widely considered as yeasts ((Saccharomyces, Candida albicans, Saitoella, Schizosaccharomyces, Sporobolomyces, see e.g. Kurtzmann et al's book The Yeasts). I do not agree with the argument that the presence of differentiated cells makes Sporobolomyces or Saitoella less exemplary yeast species. In fact, even Saccharomyces cerevisiae can produce pseudohyphae, and Schizosaccharomyces produces hyphae (see reference below). Similarly, not all CM Asco- and Basidiomycota species were not required to have a gene to call it CM-associated.

Although this is a significant difference in opinion, I respect the authors' decision to gear their analyses towards budding and fission yeast. However, in order to be transparent about potential limitations of the approach towards the readers of the paper, this decision should be made clear. Specifically, it should be mentioned in the abstract and the discussion and possibly in the methods section which yeasts and yeast-like fungi were included in the designation of CM genes and which were not. I made specific suggestions on where in the discussion this could be incorporated. I would also feel it necessary to mention that many of the >1000 genes were absent in Tuber melanosporum, a classic CM Ascomycota species. Therefore, I strongly encourage the authors to discuss the presence/absence of CM-associated genes in primarily unicellular species and CM Ascomycota (especially Tuber). This is part of the story and its inclusion will make the paper complete.

p3l2 - capitalize 'earth'?

p6l14-22 - please highlight which of these genes are related to fruiting body development? As far as I can understand the listed genes are related to hyphal fusion, which is required for fruiting body development but is not FB development per se.

p7l8 - please discuss here that they are present in Saitoella and Sporobolomyces

p11l11 - please specify Saccharomyces and Schizosaccharomyces.

p11l12-15 - it should be noted here that rudimentary CM, i.e. primitive hyphal growth is already present in earlier diverging groups such as Mucoromycotina. It would be interesting if the authors could discuss their findings in the context of the hyphal growth of even earlier diverging groups.

p13l21 - in discussing Saitoella's situation, the authors should devote a few lines to Sporobolomyces, another primarily unicellular yeast that has many of the CM-associated genes reported in the ms. In general, many yeasts are able to switch between hyphal and unicellular growth - for example Saccharomyces can produce pseudohyphae and Schizosaccharomyces can produce hyphae (Amoah-Buahin et al Eukaryotic Cell July 2005 vol. 4 no. 7 1287-1297). I feel this situation is somewhat oversimplified in the current presentation of the results and deserves more discussion. As I suggested earlier, the authors should make clear that their dataset contained 5 species generally regarded as

yeasts (*Saccharomyces*, *Candida albicans*, *Saitoella*, *Schizosaccharomyces*, *Sporobolomyces*, see Kurtzmann et al 2011 *The Yeasts*, 5th Edition) and that most of these are able to differentiate into some form of hypha-like structure, either pseudohyphae or hyphae.

With regard to my initial review, I apologize for the all capitals comment on membrane association of GFP-tagged proteins. This was a note to myself, which should have been removed from the submitted version of the review.

Reviewer #3 (Remarks to the Author):

REVIEWER #3

I have re-reviewed this interesting manuscript by Ngyuen et al. (16-17154). The authors have satisfactorily addressed each of my concerns, clarifying their arguments and strengthening an already strong piece of science.

Thank you for offering me the opportunity to serve as a reviewer for Nature Communications.

NCOMMS-16-17154A

Response to reviewer comments (highlighted in grey)

Reviewer #2 (Remarks to the Author):

Review of the revised ms by Nguyen et al on *Neolecta*. I think this is a great topic, a very interesting model and there is no doubt the ms contains several important findings. On the otherhand, I was a bit surprised by the authors' dismissal of my suggestion to include all yeasts and primarily unicellular species in their analyses. Their dataset contains 5 species that grow primarily as unicellular and are widely considered as yeasts ((*Saccharomyces*, *Candida albicans*, *Saitoella*, *Schizosaccharomyces*, *Sporobolomyces*, see e.g. Kurtzmann et al's book *The Yeasts*). I do not agree with the argument that the presence of differentiated cells makes *Sporobolomyces* or *Saitoella* less exemplary yeast species. In fact, even *Saccharomyces cerevisiae* can produce pseudohyphae, and *Schizosaccharomyces* produces hyphae (see reference below). Similarly, not all CM Asco- and Basidiomycota species were not required to have a gene to call it CM-associated.

Although this is a significant difference in opinion, I respect the authors' decision to gear their analyses towards budding and fission yeast. However, in order to be transparent about potential limitations of the approach towards the readers of the paper, this decision should be made clear. Specifically, it should be mentioned in the abstract and the discussion and possibly in the methods section which yeasts and yeast-like fungi were included in the designation of CM genes and which were not. I made specific suggestions on where in the discussion this could be incorporated. I would also feel it necessary to mention that many of the >1000 genes were absent in *Tuber melanosporum*, a classic CM Ascomycota species. Therefore, I strongly encourage the authors to discuss the presence/absence of CM-associated genes in primarily unicellular species and CM Ascomycota (especially *Tuber*). This is part of the story and its inclusion will make the paper complete.

Response concerning search strategy and choice of species

Our search strategy was prompted by the results presented in Figure 4. These data show that known developmental regulators are present in *Neolecta* and acknowledged CM taxa but not budding and fission yeasts. It can also be seen here that some of these are present in other species residing outside of acknowledged CM groups, such as *Sporobolomyces*. We were aware of this from the outset and chose to not reject genes because they hit species that do not conform to a rigid interpretation of simple versus complex. Our intention was to design a search that casts as wide a net as possible without making assumptions about the underlying complexity of less well-investigated species (For example, *Sporobolomyces* and *Saitoella*, which are both virtually unstudied). Having said this, because other readers may have a similar reaction, we have added a new paragraph in the discussion in which we explicitly discuss these issues.

“Our computational search was designed to cast a wide net for complexity-related genes and its output is not meant to definitively identify genes associated with CM. As with *Saitoella*, other

species outside of CM groups can possess genes identified by the search (e.g. *Candida albicans*). Interestingly, the choanoflagellate and filasterean sister groups to animals also do not meet the criteria of CM, yet possesses many genes associated with metazoan CM^{9,13}. The fungi present a spectrum of biological complexity whose genetic basis remains poorly understood, and many factors can contribute to the apparent presence or absence of a given gene. Thus, case-by-case experimental evidence is required to determine the function of a given gene in a given species.”

Response concerning less well-investigated species

A great deal of attention has been paid to budding and fission yeasts. For example, it is known that unipolar budding and persistent attachment of mother and daughter cells through the surface adhesin, FLO11, produce pseudohyphae in budding yeast. By contrast, we know virtually nothing about how cells are differentiated in *Sporobolomyces* or *Saitoella* nor their degree of cellular complexity. We understand why the reviewer would group these fungi based on apparent similarities. However, we believe that these species need to be more intensively investigated before conclusions concerning how they fit into the spectrum of fungal complexity can be reached. We hope that our paper stimulates the field along these lines.

Response concerning the apparent absence of CM-associated genes in Tuber

The impression that Tuber is atypically missing a large number of our identified CM-associated genes is incorrect.

Our analysis leading to the designation of CM-association included 14 Pezizomycotina species and a given gene was considered present in this subphylum only if it was detected in at least 7 species. In the initial Supplementary Table we showed 3 species that we considered representative, *Neurospora crassa* (the reference), *Aspergillus nidulans* and *Tuber melanosporum*. 39 of 1050 genes are missing in *Aspergillus*, while 55 are missing in *Tuber*. These account for 4% and 5% of the total identified genes, respectively. The impression that “many of the >1000 genes were absent in Tuber melanosporum” may be due to the clustering of these genes at the top of the table. Actually, the species with the highest level of gene absence is *Blumeria graminis* with 99. Thus, Tuber is not atypical of the other analyzed species. To avoid generating this false impression, we have expanded this Supplementary Table to show data for all 14 Pezizomycotina species. The Materials and Methods section has also been edited to clarify this point.

p3l2 - capitalize ‘earth’?

Yes, we have done this.

p6l14-22 - please highlight which of these genes are related to fruiting body development? As far as I can understand the listed genes are related to hyphal fusion, which is required for fruiting body development but is not FB development per se.

Hyphal fusion could be required for vegetative hyphae to attain competence for fruiting body development. However, it is also possible that hyphal fusion plays an essential role in both

vegetative hyphae and in the fruiting body. Because of this uncertainty, we cannot draw the distinction as suggested.

p718 - please discuss here that they are present in *Saitoella* and *Sporobolomyces*

We feel that this is tangential to our narrative. However, we have added a paragraph to the discussion (see above), which explicitly discusses the limitations of a purely computational approach.

p11111 - please specify *Saccharomyces* and *Schizosaccharomyces*.

Yes, we have done this. Note that we use the term budding and fission yeast, which are defined for these species in the last paragraph of the introduction.

p11112-15 - it should be noted here that rudimentary CM, i.e. primitive hyphal growth is already present in earlier diverging groups such as *Mucoromycotina*. It would be interesting if the authors could discuss their findings in the context of the hyphal growth of even earlier diverging groups.

Neolecta's cellular and morphological similarity to the *Pezizomycotina* combined with its overlapping genetic repertoire of CM-related genes warrants the statement that "rudimentary CM is deeply rooted in the *Ascomycota*". *Neolecta* and the acknowledged CM groups *Pezizomycotina* and *Agaricomycotina* all produce perforate septa with pore-gating mechanisms. This is not the case with the early-diverging fungi (e.g. *Mucoromycotina*). In addition, these fungi do not produce multicellular fruiting bodies. It is therefore unclear why the reviewer uses the presence of simple hyphae to justify the assignment of "rudimentary CM". While we agree that a discussion of the early diverging fungi is in principle interesting, we feel that this is tangential to the focus of this paper.

p13121 - in discussing *Saitoella*'s situation, the authors should devote a few lines to *Sporobolomyces*, another primarily unicellular yeast that has many of the CM-associated genes reported in the ms. In general, many yeasts are able to switch between hyphal and unicellular growth - for example *Saccharomyces* can produce pseudohyphae and *Schizosaccharomyces* can produce hyphae (Amoah-Buahin et al *Eukaryotic Cell* July 2005 vol. 4 no. 7 1287-1297). I feel this situation is somewhat oversimplified in the current presentation of the results and deserves more discussion. As I suggested earlier, the authors should make clear that their dataset contained 5 species generally regarded as yeasts (*Saccharomyces*, *Candida albicans*, *Saitoella*, *Schizosaccharomyces*, *Sporobolomyces*, see Kurtzmann et al 2011 *The Yeasts*, 5th Edition) and that most of these are able to differentiate into some form of hypha-like structure, either pseudohyphae or hyphae.

Please refer to the blue highlighted paragraph above for the response to a similar point. We wanted to construct a paper that would be accessible to people outside the fungal field and therefore do not see the point of interjecting a discussion of *Sporobolomyces*, which is not in the

Ascomycota. We feel that the discussion on *Saitoella* combined with the new paragraph discussing the limitation of the computation search fully address the concern raised by this comment.

With regard to my initial review, I apologize for the all capitals comment on membrane association of GFP-tagged proteins. This was a note to myself, which should have been removed from the submitted version of the review.